# Plasma electron acceleration driven by a long-wave-infrared laser

R. Zgadzaj[1], J. Welch[1], Y. Cao [ORCID][1], L. D. Amorim [ORCID][2], A. Cheng[2], A. Gaikwad[2], P. Iapozzutto[2], P. Kumar [ORCID][2], V. N. Litvinenko [ORCID][2], I. Petrushina[2], R. Samulyak[2], N. Vafaei-Najafabadi [ORCID][2], C. Joshi[3], C. Zhang [ORCID][3], M. Babzien[4], M. Fedurin[4], R. Kupfer[4], K. Kusche[4], M. A. Palmer [ORCID][4], I. V. Pogorelsky[4], M. N. Polyanskiy [ORCID][4], C. Swinson[4] & M. C. Downer [ORCID][1] ✉

Laser-driven plasma accelerators provide tabletop sources of relativistic electron bunches and femtosecond x-ray pulses, but usually require petawatt-class solid-state-laser pulses of wavelength $\lambda_L \sim 1\,\mu m$. Longer-$\lambda_L$ lasers can potentially accelerate higher-quality bunches, since they require less power to drive larger wakes in less dense plasma. Here, we report on a self-injecting plasma accelerator driven by a long-wave-infrared laser: a chirped-pulse-amplified $CO_2$ laser ($\lambda_L \approx 10\,\mu m$). Through optical scattering experiments, we observed wakes that 4-ps $CO_2$ pulses with < 1/2 terawatt (TW) peak power drove in hydrogen plasma of electron density down to $4 \times 10^{17}\,cm^{-3}$ (1/100 atmospheric density) via a self-modulation (SM) instability. Shorter, more powerful $CO_2$ pulses drove wakes in plasma down to $3 \times 10^{16}\,cm^{-3}$ that captured and accelerated plasma electrons to relativistic energy. Collimated quasi-monoenergetic features in the electron output marked the onset of a transition from SM to bubble-regime acceleration, portending future higher-quality accelerators driven by yet shorter, more powerful pulses.

Since Tajima and Dawson proposed the idea of accelerating charged particles by surfing them on light-speed plasma waves[1], plasma-based wakefield accelerators (WFAs) have fueled a worldwide quest for more compact, less expensive alternatives to conventional radio-frequency (rf) accelerators[2,3]. Tabletop laser-driven WFAs (LWFAs) have accelerated high-quality electron bunches to nearly 10 GeV within a few centimeters[4]. LWFAs underlie femtosecond X-ray sources[5], and are part of mainstream planning for 21st century accelerator science in the U.S.[6], Europe[7] and the U.K.[8]. Chirped-pulse amplified (CPA) lasers[9] that produce light pulses powerful and short enough to drive the high-amplitude, near-light-speed plasma waves needed to capture and accelerate electrons drove LWFA development for a quarter-century. But only solid-state CPA lasers at wavelengths $\lambda_L \sim 1\,\mu m$ have done so, except for a recent demonstration of an LWFA driven by a mid-wave-infrared laser ($\lambda_L = 3.9\,\mu m$)[10]. Long-wave-infrared (LWIR, $8 < \lambda_L < 15\,\mu m$) CPA lasers producing pulses with terawatt peak power[11,12] open new

opportunities for LWFAs[13]. Here, we demonstrate an LWFA driven by picosecond $\lambda_L \sim 10\,\mu m$ pulses from a CPA carbon-dioxide ($CO_2$) laser, and diagnose the properties of the laser-driven plasma waves and relativistic electrons that these waves capture and accelerate.

In their original proposal[1], Tajima and Dawson envisioned a laser (L) pulse of duration $\tau_L < \pi/\omega_p$ impulsively exciting a collective electron-density (Langmuir) wave at the natural plasma frequency $\omega_p = [n_e e^2/\epsilon_0 m_e]^{1/2}$. Here, $n_e$ is the plasma's unperturbed electron density, $e$ and $m_e$ denote electron charge and mass, respectively, and $\epsilon_0$ is the permittivity of free space. Such a pulse launches a plasma wave resonantly by expelling plasma electrons from within its sub-period envelope by exerting ponderomotive pressure[14], equivalent to the gradient $\nabla(\epsilon_0 E_L^2/2)$ of the pulse's cycle-averaged electromagnetic energy density, where $E_L$ is the optical field strength. To drive waves to their full amplitude, the pulse must impart relativistic momentum $eE_L/\omega_L \gtrsim m_e c$ to plasma electrons within each optical cycle $\omega_L^{-1}$, i.e. the

[1]University of Texas at Austin, 2515 Speedway C1600, Austin, TX 78712, USA. [2]Stony Brook University, Stony Brook, NY 11794, USA. [3]University of California at Los Angeles, Los Angeles, CA 90024, USA. [4]Brookhaven National Laboratory, Upton, NY 11973, USA. ✉e-mail: downer@physics.utexas.edu

momentum ratio $a_0 \equiv eE_L/\omega_L m_e c$, often called the dimensionless field amplitude or normalized vector potential, must exceed 1. This in turn necessitates peak intensity $I_L$ [W/cm²] $\gtrsim (a_0/\lambda_L[\mu m])^2 \times 10^{18}$, and yields longitudinal electrostatic fields $E_z$ [V/cm] $\approx (n_e$ [cm⁻³]$)^{1/2}$ within the driven waves. In order for $E_z$ to exceed accelerating fields in conventional rf accelerators ($\sim 10^6$ V/cm) by an interesting factor of $\gtrsim 10^2$, plasma of density $n_e \gtrsim 10^{16}$ cm⁻³, and drive pulses of duration $\tau_L \lesssim 1$ ps with $I_L \gtrsim (a_0/\lambda_L[\mu m])^2 \times 10^{18}$ W/cm² are needed. While today's $\lambda_L \sim 1\mu m$ CPA lasers routinely provide such pulses, CPA lasers available in the 1990s did not.

Instead, researchers at that time discovered two alternative LWFA drive schemes that circumvented these requirements. In one, the drive laser (coincidentally $CO_2$) operated at two closely-spaced frequencies whose beat frequency matched $\omega_p$ of $n_e \approx 10^{16}$ cm⁻³ plasma. The dual-wavelength pulses thus drove plasma waves resonantly to high-amplitude at this specific density, and accelerated small numbers of externally-injected electrons, despite its sub-relativistic $I_L$ and multi-$\lambda_p/c$ duration[15,16]. In the second scheme, researchers focused CPA pulses with $\lambda_L \sim 1\mu m$ and $\tau_L \gg \lambda_p/c$ into near atmospheric density plasma ($n_e \sim 10^{19}$ cm⁻³), since shorter duration pulses were not yet available. Nevertheless, strong wakes were generated and copious self-injected tens-of-MeV electrons produced[17] when the peak power $P_L$ of the drive pulse exceeded the critical power[18,19]

$$P_{cr}[\text{TW}] = 0.017 \frac{n_{cr}}{n_e} = \frac{2000}{n_e[10^{16} \text{ cm}^{-3}](\lambda_L[\mu m])^2} \quad (1)$$

for relativistic self-focusing (RSF), which is favored at high $n_e$. Here, $n_{cr} = \epsilon_0 m_e \omega_L^2/e^2 = (1.1 \times 10^{21})/(\lambda_L[\mu m])^2$ cm⁻³ is the critical plasma density at which $\omega_L = \omega_p$. Because of RSF, the drive pulse reached, and self-guided at, higher $a_0$ inside the plasma than it had upon entering the plasma, enabling it to drive forward Raman instabilities[18–20]. These instabilities broke up the pulse into a train of sub-pulses spaced by $\lambda_p$, each of length $c\tau_L \lesssim \lambda_p$ and relativistic strength $a_0 \gtrsim 1$. Consequently they drove a wake beyond the wave-breaking limit, triggering self-injection of plasma electrons, as Stokes and anti-Stokes sidebands at $\pm n\omega_p(n=1,2,3,\ldots)$ appeared on the transmitted drive pulse spectrum. Although these self-modulated (SM) LWFAs yielded electron bunches of lower energy, and wider energy and angular spread, than today's impulsively-excited bubble-regime LWFAs[21], their decade-long (1995-2004) study uncovered much LWFA physics relevant to the latter regime, and drove short-pulse CPA technology needed to realize it. Moreover, SM-LWFAs remain of contemporary interest as strong betatron x-ray emitters[22] and as models for self-modulated proton-driven plasma-based accelerators[23].

For LWFA applications, today's CPA $CO_2$ lasers (pulse energy $\mathcal{E}_L \lesssim 10$ J, duration $\tau_L \approx 2$ ps[12]) have developed to a stage analogous to 1990s-era 1-$\mu m$ CPA lasers. The duration of their shortest pulses still exceeds an oscillation period ($\lambda_p/c \approx 1$ ps) of $n_e = 10^{16}$ cm⁻³ plasma, preventing bubble-regime excitation. Nevertheless, simulations[24] indicate that the bubble regime is within reach with only 4 (2.5)-fold improvement in $\tau_L$ ($\mathcal{E}_L$), offering the prospect of bubble structures of unprecedented size $\lambda_p \approx 300\mu m$, along with better control of LWFA and higher e-beam quality. Meanwhile, $\sim 10$-$\mu m$ CPA pulses available here provided nominally $P_L \approx \mathcal{E}_L/\tau_L \approx 2$ TW, which exceeds $P_{cr}$ for $n_e$ as low as $10^{17}$ cm⁻³ [see Eq. (1)]. This enables SM-LWFA at $> 100 \times$ lower $n_e$, via wakes of $10 \times$ larger $\lambda_p$, than was possible with 1-$\mu m$ CPA lasers of equivalent $P_L$ or demonstrated with mid-wave-infrared CPA lasers[10]. Equivalently, at fixed $n_e$ current 10-$\mu m$ CPA pulses can trigger SM-LWFA at $100 \times$ lower $P_L$ than 1-$\mu m$ pulses: e.g., a recent study of SM-LWFA at $n_e = 3 \times 10^{17}$ cm⁻³ used 1-$\mu m$ pulses of $P_L \approx 170$ TW[25]. Simulations[26,27] have borne out these general expectations for $\sim 10$-$\mu m$ CPA pulses. Here, we demonstrate them in the laboratory for the first time. We first characterize SM-LWFA structures generated by 2 J, 4 ps pulses at $P_L > P_{cr}$, but below the threshold of electron self-injection.

Then, following a laser upgrade nominally to $\sim$ 4 J, 2 ps pulses, with occasional more powerful pulses available, we characterized MeV electrons generated at $P_L > P_{cr}$. But unexpectedly, we also observed electrons for $P_L$ as low as $0.3P_{cr}$ (i.e. $n_e$ down to $3 \times 10^{16}$ cm⁻³), indicating that self-focusing was no longer essential to exciting plasma wakes or to capturing and accelerating plasma electrons. Moreover, collimated, quasi-monoenergetic electrons accompanied the divergent, thermal electrons traditionally generated by SM-LWFA, indicating that we had entered a transitional LWFA regime intermediate between SM and bubble-regime LWFA[28]. The results thus represent a steppingstone toward bubble-regime LWFAs of unprecedented spatial scale in $n_e \sim 10^{16}$ cm⁻³ plasma, which offer the possibilities of precisely injecting synchronized low-energy-spread, low-emittance bunches from conventional linacs into LWFAs. Large bubbles in turn offer excellent prospects for preserving high beam quality during acceleration, and thus for driving the next generation's coherent X-ray sources[29].

## Results

### Generation of self-modulated wakes

Experiments were carried out at Brookhaven National Laboratory's (BNL's) Accelerator Test Facility (ATF)[30]. To generate SM wakes, an off-axis parabola (OAP) mirror focused linearly-polarized drive pulses from ATF's CPA $CO_2$ laser[11,12] to Gaussian spot radius $w_0 \approx 27.5 \mu m$ at a focal plane located $1 \pm 0.1$ mm before the center of a supersonic hydrogen gas jet with an axially symmetric profile of 2 mm diameter. The dashed curve in Fig. 1a shows an idealized electron density profile $n_e(z)$ of the ionized gas jet along the laser propagation axis, here with plateau density $n_e = 5 \times 10^{17}$ cm⁻³, that we used for simulations. In simulations the laser focal plane was at $z = 0.1$ mm, near the beginning of the density plateau (see Methods/Simulations for further discussion). The laser focal spot matched half a plasma wavelength $\lambda_p/2 = \pi/k_p$ (where $k_p = 2\pi/\lambda_p$ is the plasma wavenumber) for plateau density $n_e = 4 \times 10^{17}$ cm⁻³, and thus satisfied a transverse near-resonant excitation condition $k_p w_0 \sim \pi$ to within a factor of two over the density range $10^{17} \lesssim n_e \lesssim 10^{18}$ cm⁻³ of interest here, even though the pulses were mismatched to the longitudinal resonant condition $\omega_p \tau_L \sim \pi$ by factors ranging from 7 (for $n_e = 4 \times 10^{16}$ cm⁻³, $\tau_L = 2$ ps) to 100 (for $n_e = 2 \times 10^{18}$ cm⁻³, $\tau_L = 4$ ps). This contributed to efficient excitation of stable longitudinally-propagating plasma waves, and contrasts with most previous SM-LWFA experiments[17], in which $\lambda_L \approx 1\mu m$ drive pulses were focused to spot sizes $w_0 \gg \lambda_p/2$ well outside the transverse resonant condition, subjecting the drive pulse to filamentation.

To generate wakes that did not capture and accelerate electrons, the ATF laser delivered drive pulses of 4 ps (FWHM) duration and up to 2 J energy (i.e. $P_L \lesssim 0.5$ TW) at wavelength $\lambda_L = 10.3 \mu m$. Focused pulses thus had peak vacuum intensity up to $I_0 \approx 4 \times 10^{16}$ W/cm², or vacuum laser strength parameter $a_0^{(vac)} \approx 1.8$. Since here $a_0 \gtrsim 1$, the interaction was mildly relativistic. Simulations of the interaction that include tunneling ionization, exemplified by Fig. 1b–c, show that under these conditions, the wings of a focused Gaussian $CO_2$ laser pulse self-ionize a hydrogen column of radius $R_p \sim 50 \mu m$ (see Fig. 1b). Non-Gaussian (e.g. Lorentzian, aberrated) pulses of the same FWHM would ionize an even wider column because of their more intense wings. Regardless of the exact $R_p$, the drive pulse generated wakes of transverse radius $R_w \approx w_0 \approx 20 \mu m$ that lay entirely within the self-ionized plasma column. Thus pre-ionization was not essential to produce a plasma wide enough to support the wake oscillations. By adjusting backing pressure of the gas jet nozzle, electron density $n_e$ of the resulting plasma was varied over the range $10^{17}$ cm⁻³ $< n_e < 2 \times 10^{18}$ cm⁻³, which corresponded to a range $1.7 > P_{cr} > 0.08$ TW of critical powers [see Eq. (1)] that straddled the maximum available incident peak power $P_L^{(max)} \approx 0.5$ TW. Experiments discussed below detected wakes only for $P_L > P_{cr}$, i.e. for $n_e > 3.5 \times 10^{17}$ cm⁻³. Their wavelengths $\lambda_p$ ranged from 56 $\mu m$ at this threshold to 24 $\mu m$ at $n_e = 2 \times 10^{18}$ cm⁻³. This threshold $n_e$ was nearly $100 \times$ lower than densities at which $\lambda_L = 1\mu m$ laser pulses of similar $P_L$

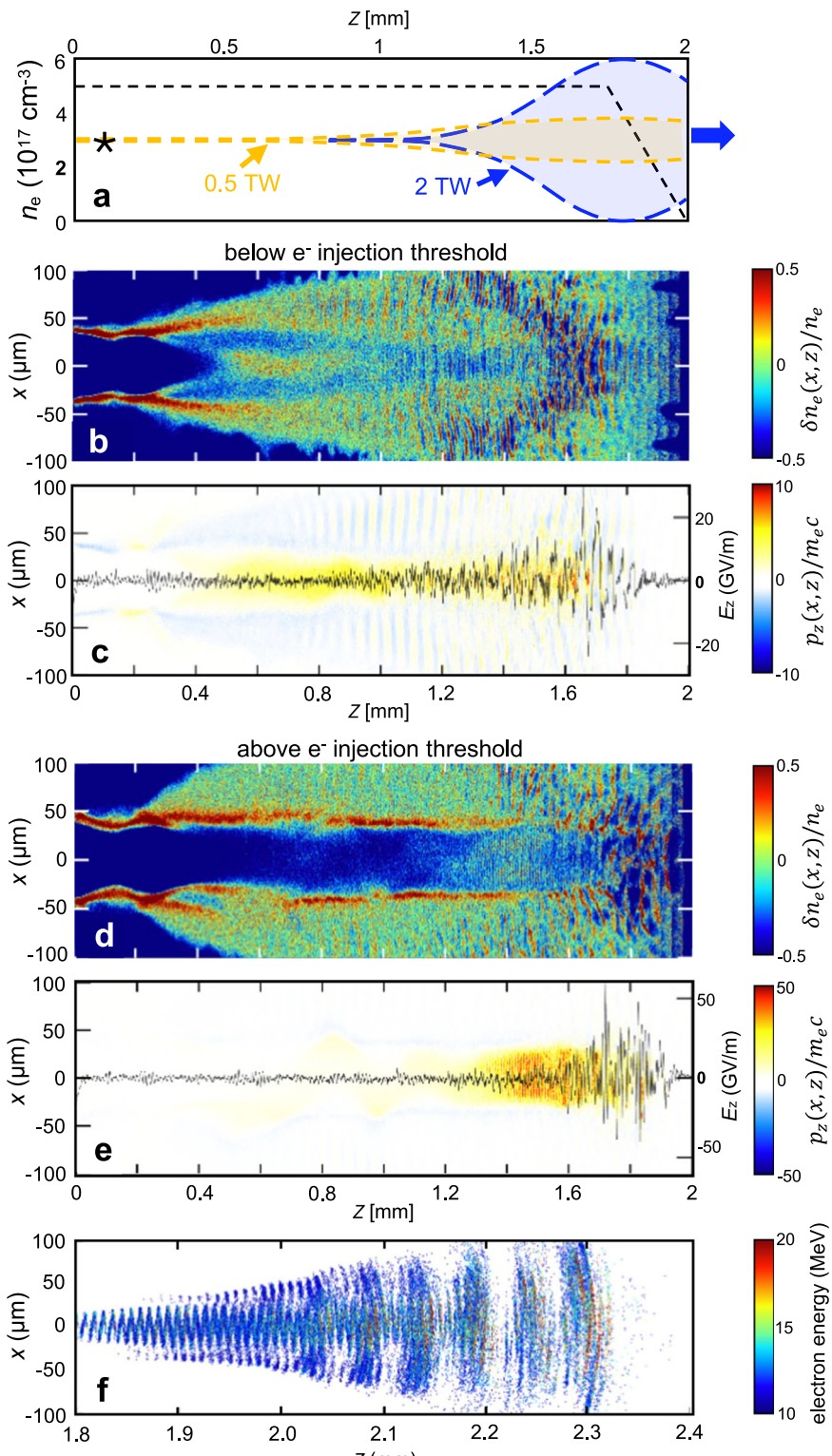

**Fig. 1 | 3D particle-in-cell simulations of SM-LWFA. a** Electron density $n_e(z)$ profile (black dashed curve) of unperturbed, fully-ionized 2 mm gas jet with $n_e = 5 \times 10^{17}$ cm$^{-3}$ plateau and 0.25 mm exit ramp used for simulations. Vacuum field envelopes of right-propagating 2 J, 4 ps (0.5 TW, yellow) and 4 J, 2 ps (2 TW, blue) pulses, which vacuum-focused at $z = 0.1$ mm (marked by star), are shown at $z = 1.7$ mm. Remaining panels: 2D wake profiles $\delta n_e(x,z)/n_e$ when **b** 0.5 TW or **d** 2 TW pulses reach $z = 1.7$ mm, and simultaneous normalized 2D electron momentum profiles $p_z(x,z)/m_e c$ for **c** 0.5 TW and **e** 2 TW excitation, i.e. below and above self-injection threshold, respectively. Black curves in (**c**) and (**e**): longitudinal electric field $E_z(z)$ profiles on the laser propagation axis, referenced to right-hand vertical scales. **f** Projection onto a plane of ≥10 MeV electrons at a later instant, after the 2-TW-laser-driven wake accelerated them into vacuum.

generated detectable SM wakes[17]. Panels b and c of Fig. 1 show simulated temporal snapshots of the electron density $n_e(x, z)$ (b) and longitudinal momentum $p_z(x, z)$ (c) profiles of wake oscillations for plateau density $n_e = 5 \times 10^{17}$ cm$^{-3}$, after the $\lambda_L = 10$ $\mu$m, 0.5 TW drive pulse propagated to the end ($z = 1.6$ mm) of the gas jet's density plateau (see Methods for details of simulations). Our simulations presented in ref. 27 show a strong influence of the dynamic ionization model on the structure and evolution of the SM wakes compared to the pre-ionized plasma approximation. In dynamically ionized gas the laser pulse modulates more strongly, and stronger wakes form earlier because of the stronger ponderomotive force. Likewise, including ion motion in simulations leads to the formation of ionization channels (Fig. 1b,d), which are largely suppressed in simulations using the fixed ion approximation, as shown in Supplementary Fig. 1. ref. 27 further discusses the role of dynamic ionization and ion motion in wake formation.

Even though wakes formed for $P_L > P_{cr}$, the $p_z(x, z)$ profile in Fig. 1c shows only momenta attributable to the wake oscillations themselves. The additional $z$-momentum that would be expected if the wake had trapped and accelerated electrons is not present. This means that at $P_L = 0.5$ TW and $n_e = 5 \times 10^{17}$ cm$^{-3}$ we are above the self-focusing threshold needed to form wake structures, but below the self-injection threshold. This absence of self-injected, trapped electrons accelerating along the wake propagation direction in these simulations corroborates our observation of no accelerated electrons.

To generate strongly nonlinear SM wakes that captured and accelerated plasma electrons to produce a collimated beam, the $CO_2$ laser was upgraded (see Methods) to deliver nominally 2 ps, $\lambda_L = 9.2$ $\mu$m pulses with energy up to 4 J to the jet with the same focus. Occasional individual pulses with $\tau_L$ as small as 1.8 ps, $\mathcal{E}_L$ has high as 6 J, and $P_L$ approaching ~ 3 TW, with only slightly degraded focus (see Methods), were measured at the vacuum interaction region. With the upgraded pulses, we observed relativistic electron production at densities down to $n_e \approx 3 \times 10^{16}$ cm$^{-3}$. Vacuum peak intensity now reached $I_0 \approx 2.5 \times 10^{17}$ W/cm$^2$ ($a_0 \approx 3.9$) for 6 J pulses. As a result, the interaction became strongly relativistic, and the forward Raman instability grew more rapidly than for the 0.5 TW pump. Panels d and e of Fig. 1 show simulations of the corresponding $n_e(x, z)$ (d) and $p_z(x, z)$ (e) profiles of SM wakes that 2 TW pulses drive in self-ionized plasma of plateau density $n_e = 5 \times 10^{17}$ cm$^{-3}$. Compared to the $n_e(x, z)$ profile in Fig. 1b, the self-ionized plasma column is twice as wide, relativistic self-focusing is stronger and wake oscillations reach wave-breaking amplitude (see Fig. 1d). In contrast to the $p_z(x, z)$ profile in Fig. 1c, copious electrons with relativistic $p_z$ are now evident. In Fig. 1e, electron bunches accelerated to $p_z/m_e c$ ~ 40 (red) are distributed among the multiple accelerating bins of the wake that the drive pulse overlapped. Moreover they are distributed randomly throughout each bin because the plasma waves had broken, injecting plasma electrons at uncontrolled initial locations and times prior to their trapping in the wake's accelerating potential. This wave-breaking and injection, once started in mid-jet, continued through the end of the interaction, since they are the culmination of the forward Raman instability. Figure 1f shows a subset of the accelerated electrons with $E_e > 10$ MeV after they had propagated into vacuum. The simulated angular and energy distributions of these electrons are shown later in comparison with experimental data.

## Characterization of self-modulated wakes

Figure 2 a shows the schematic setup for probing wakes generated under the conditions of Fig. 1b,c via forward collective Thomson scatter (CTS)[31,32]. When the $CO_2$ pump drove wake oscillations $\delta n_e(\mathbf{x}, t)$ above the level of thermal fluctuations, they appeared to a green ($\lambda_{pr} = 0.532$ $\mu$m) co-propagating probe pulse of duration $\tau_{pr} = 4$ ps $> \omega_p^{-1}$ (see Methods) as a refractive index grating $\delta\eta(\mathbf{x}, t) \approx \delta n_e(\mathbf{x}, t)/2n_{cr}$ moving at phase velocity $\omega_p/k_p$[33]. This grating scattered probe light at

frequencies $\omega_{pr} \pm \omega_p$ and wave vectors $\mathbf{k}_{pr} \pm \mathbf{k}_p$, over and above background Thomson scatter at $\omega_{pr}$ from uncorrelated individual electrons. A lens (not shown) collected forward CTS probe light, and relayed it to the entrance slit of a spectrometer through a notch interference filter that blocked frequencies within the bandwidth of the incident probe, but transmitted its Stokes and anti-Stokes sidebands. When the delay $\Delta t$ between the green probe and $CO_2$ pulses was 0, the overlapped pump and probe pulses also generated difference- and sum-frequency signals at $\omega_{pr} \pm \omega_L$. The raw spectrometer data in Fig. 2b, taken at $\Delta t = 0$, shows both Stokes/anti-Stokes and difference-/sum-frequency generation (DFG/SFG) signals for seven different values of plasma density in the range $0.57 \leq n_e \leq 1.69 \times 10^{18}$ cm$^{-3}$, calibrated by independent optical measurements of the density profile $n_e(z)$ of the ionized gas jet with $\pm 10\%$ accuracy using an ionization-induced plasma grating technique[34]. The magnitude $\omega_p \sim n_e^{1/2}$ of the Stokes/anti-Stokes shifts increased as expected with $n_e$, whereas the DFG/SFG peaks remained at $n_e$-independent frequencies and helped to calibrate the spectrometer's frequency scale.

Figure 3 a compares the seven measured Stokes and anti-Stokes shifts $|\Delta\lambda_{CTS}(n_e)|$ (data points) from Fig. 2b quantitatively with $\lambda_p(n_e)$. The right-hand vertical scale gives the corresponding frequency shifts $\Delta f_{CTS}(n_e)$. The agreement is excellent. Because of the low $n_e$, these frequency shifts are much smaller $- 7 < |\Delta\lambda_{CTS}| < 12$ nm $-$ than those observed by LeBlanc et al.[35] $-$ namely $|\Delta\lambda_{CTS}| = 45$ nm, $|\Delta f_{CTS}(n_e)| = 48$ THz $-$ by probing SM wakes driven by $\lambda_L = 1$ $\mu$m pulses in $n_e = 3 \times 10^{19}$ cm$^{-3}$ plasma using the same $\lambda_{pr}$ (see e.g. Fig. 1 of ref. 35). Moreover, in the previous experiment DFG and SFG peaks could not be observed because their shift from $\lambda_{pr}$ was beyond the range of the CTS spectrometer. The green-shaded region of Fig. 3a corresponds to Stokes/anti-Stokes shifts for $n_e < 4 \times 10^{17}$ cm$^{-3}$, but was notch-filtered. Nevertheless, by rotating this interference filter slightly we could leak light from the blocked spectral window $525 < \lambda < 537$ nm into the CTS spectrometer. Through such measurements we confirmed, as shown in Fig. 3b, that sideband intensity within this window, was vanishingly weak or absent for $n_e < 4 \times 10^{17}$ cm$^{-3}$ for excitation at $P_L$. From Eq. (1), this turn-on density corresponds to $P_{cr} = P_L = 0.5$ TW. The red-shaded region in Fig. 3a corresponds to $n_e > 2 \times 10^{18}$ cm$^{-3}$, i.e. densities within a factor of 5 of the critical density for $\lambda_L = 10$ $\mu$m light. At these densities, back-reflections of the incident $CO_2$ laser light from the gas jet became strong enough to endanger upstream optics in the $CO_2$ laser system. We therefore avoided densities in this range.

The data points in Fig. 3b illustrate how side-band intensity varied as $n_e$ increased from the lower to upper threshold described above, with $P_L$ fixed at 0.5 TW and $\Delta t \approx 0$. Each data point represents an average over several shots and over the Stokes and anti-Stokes sidebands. Sideband intensity rose sharply as $n_e$ increased from $4 \times 10^{17}$ cm$^{-3}$ to $7 \times 10^{17}$ cm$^{-3}$, then fell off equally sharply at higher $n_e$. There is no single explanation for this trend. The main factors governing sideband intensity are: *i)* wake amplitude; *ii)* wake lifetime within the 4 ps probe longitudinal envelope; *iii)* wake location within that envelope; *iv)* dephasing between co-propagating wake and probe. Generally, strong sidebands occur when a high-amplitude wake filling a large fraction of the most intense portion of probe's longitudinal envelope co-propagates with the probe for approximately one coherence length[31]. As $n_e$ increased from zero, wakes began to form when $P_L \approx P_{cr}$, here at $n_e \approx 4 \times 10^{17}$ cm$^{-3}$. As $n_e$ increased further, $P_{cr}$ decreased, causing stronger self-focusing near the jet exit. Initially, this simply increased wake amplitude, causing stronger CTS. Eventually, however, the pump over-focused, generating wakes that decayed and become chaotic faster, and formed earlier both within the probe profile, where intensity was weaker, and within the jet, resulting in stronger dephasing. These factors combined to weaken CTS. These results emphasize that sideband intensity is not simply proportional to wake amplitude.

To emulate the observed trend in probe CTS intensity at minimal computation cost, we simulated the corresponding trend in pump

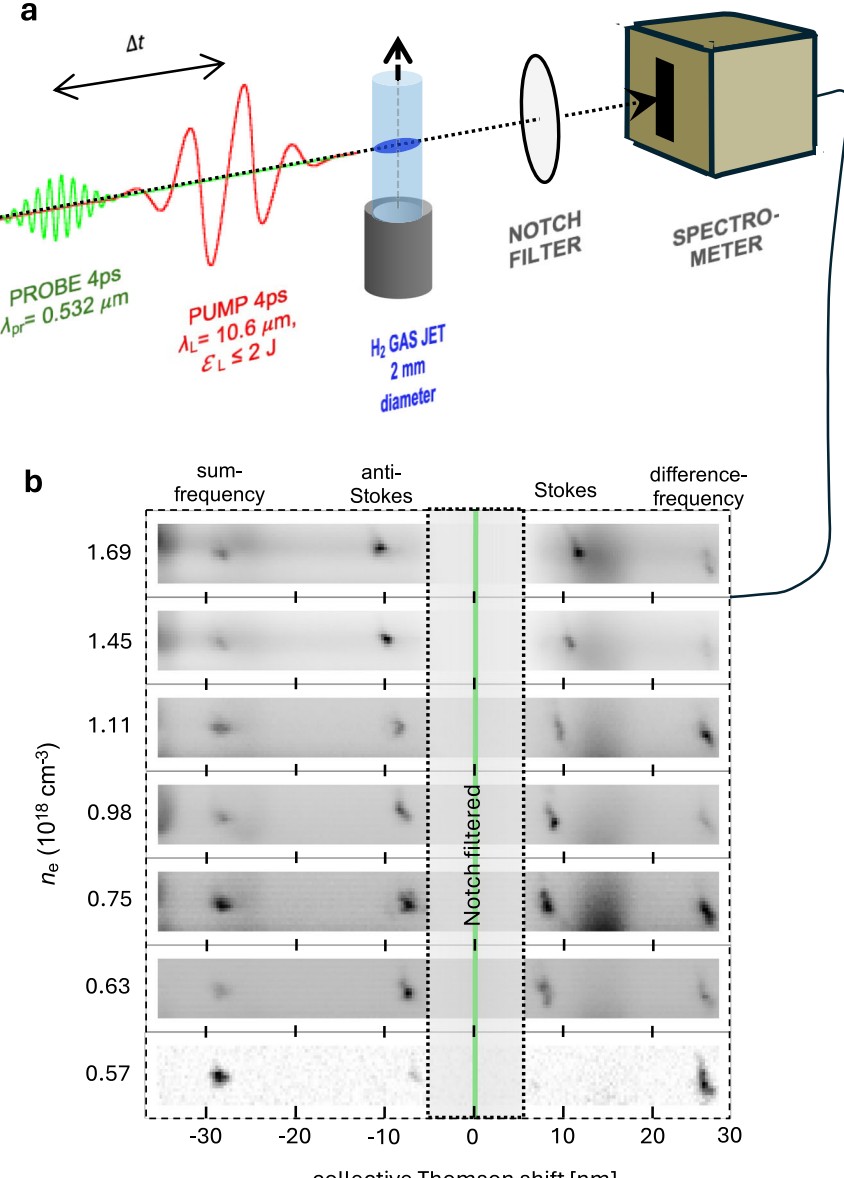

**Fig. 2 | Collective Thomson Scatter probing of SM wakes. a** Schematic experimental setup with green probe pulse co-propagating at delay $\Delta t$ behind $CO_2$ pump pulse. **b** Spectra of forward-scattered probe light for $P_L = 0.5$ TW and $\Delta t \approx 0$, for seven indicated plasma densities $n_e$, showing $n_e$-dependent anti-Stokes/Stokes sidebands due to CTS from wake and $n_e$-independent difference- and sum-frequency-generation (DFG/SFG) peaks at $\omega_{pr} \pm \omega_L$.

CTS, which requires tracking only a single light frequency. Qualitatively similar $n_e$-dependence is expected, since the pump coincided temporally with the probe at $\Delta t \approx 0$. We therefore self-consistently simulated wake formation/propagation and pump spectral evolution at four $n_e$ within the measured range[27]. Blue and red data points in Fig. 3b show the results. We applied an overall vertical shift to simulated pump Stokes (blue) and anti-Stokes (red) intensities to optimize the fit to relative probe sidebands, since absolute sideband intensities could not be measured accurately. The simulations qualitatively reproduced the observed trend of an initial increase in sideband intensity followed by a decrease at higher density.

Figure 4 a illustrates how the intensity and spectral shape of the first Stokes sideband varied as $P_L/P_{cr}$ increased from ~ 0.2 to 2.5 at fixed density $n_e = 1.3 \times 10^{18}$ cm$^{-3}$ and delay $\Delta t \approx 0$. Spectra of the probe pulse ($\lambda_{pr} = 0.532\,\mu$m), a small portion of which was leaked into the spectrometer by rotating the notch interference filters, and of the difference-frequency signal ($\lambda_{DFG} \approx 0.560\,\mu$m) are shown for comparison. For $P_L/P_{cr} < 1$, no sidebands were observed. Then within the narrow range

$1 < P_L/P_{cr} < 1.2$ sideband intensity grew quickly. For $P_L/P_{cr} > 1.2$ it fluctuated erratically from shot-to-shot around an average, but nearly $P_L$-independent, value. The solid red curve in Fig. 4a summarizes these trends, which mirrored those of the anti-Stokes sideband. DFG signals were observed at normalized power down to $P_L/P_{cr} \approx 0.5$, then strengthened gradually with increasing $P_L$. RMS fluctuations of both CTS sideband and DFG signals significantly exceeded those of the probe power $P_{pr}$ itself, as is evident from the leaked probe signals in Fig. 4a.

To understand these trends, we must take into account the influence of wave vector mismatch on both forward CTS and DFG/SFG signals. Forward-scattered power $P_{S,AS}$ of first-order Stokes (S) or anti-Stokes (AS) sidebands, normalized to the portion of probe power $P_{pr}$ that overlaps the wake of amplitude $\delta n_e$ over interaction length $L$, is given by[31,32]

$$\frac{P_{S,AS}}{P_{pr}} = \frac{1}{4}(\delta n_e)^2 r_0^2 \lambda_{pr}^2 L^2 F, \qquad (2)$$

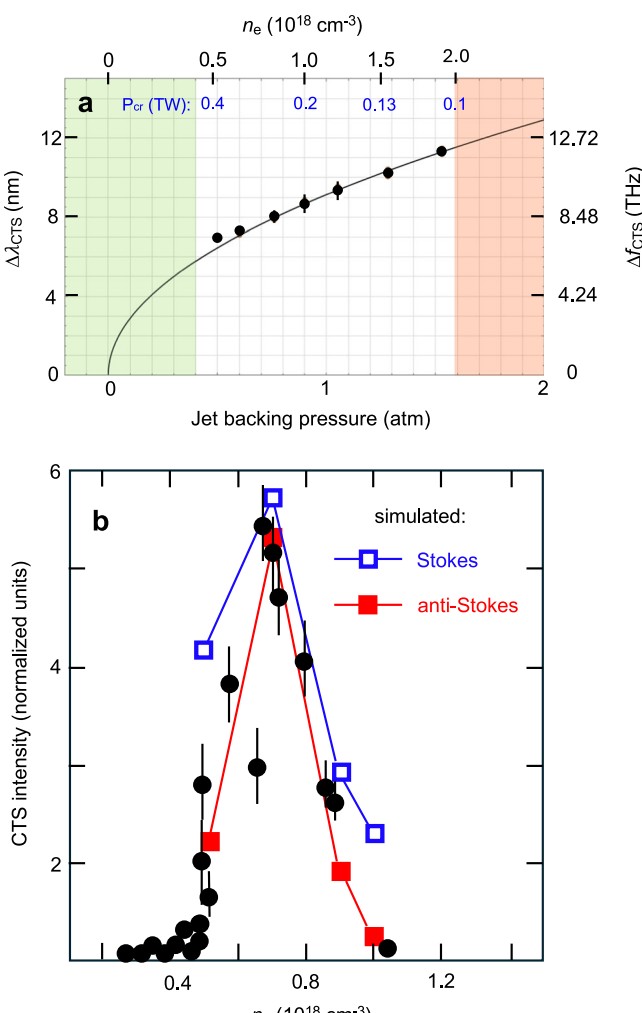

**Fig. 3 | Quantitative $n_e$-dependence of CTS. a** Spectral Stokes/anti-Stokes shift vs. $n_e$. Data points: observed shifts for the seven $n_e$ values in Fig. 2b. Solid curve: plasma wavelength vs. $n_e$. Green shading: no sidebands observed. Red shading: no measurements attempted because of strong pump back-reflections from ionized gas jet. **b** Probe sideband intensity vs. $n_e$ for $P_L = 0.5$ TW and $\Delta t \approx 0$. Black data points: measured average probe Stokes/anti-Stokes intensities. Colored data points and connecting lines: simulated pump Stokes (blue) and anti-Stokes (orange) sideband intensities at $n_e = 5, 7, 9$ and $10 \times 10^{17}$ cm$^{-3}$. Error bars in both **a** and **b**: 1 standard deviation of variation among repeated runs.

where $r_0$ is the classical electron radius, $F \equiv \sin^2(\Delta kL)/(\Delta kL)^2$ is the wave vector mismatch factor and $\Delta k = k_{pr} - k_{S,AS} \pm k_p$ is the mismatch of the $z$-components of the wave vectors. Here, $k_{pr}$ is the $z$-component of the wave vector of incident probe light, $k_{S,AS}$ of forward-scattered sidebands and $k_p = \omega_p/v_p$ of the plasma wave, the phase velocity $v_p$ of which equals pump group velocity $v_g$, and the sign of which is taken as $+ (-)$ for S (AS). For the conditions of Fig. 4a, one finds $\Delta k \approx 150$ cm$^{-1}$, implying coherence length $L_{coh} = \pi/(4|\Delta k|) \approx 50 \mu m$, i.e. CTS signals grow only over propagation distances ~50 $\mu m$, < 5% of the plasma density plateau length (see Fig. 1a), before de-phasing from, and destructively interfering with, previously generated CTS light. Thus fluctuations in $L$ of only 50 $\mu m$, which we cannot control, can cause $P_s$ to fluctuate between 0 and its maximum value. This explains why shot-to-shot fluctuations in $P_{S,AS}$ exceed those in $P_{pr}$. A factor of the same form as $F$, with $\Delta k' = k_{DFG/SFG} - k_{pr} \pm k_{pu}$ replacing $\Delta k$, governs DFG/SFG. One finds $L_{coh}^{DFG/SFG} \approx 10 \mu m$, implying even greater sensitivity to fluctuations in $L$.

Normalized wake amplitude $\delta n_e/n_e$ was difficult to estimate accurately from Eq. (2), since an absolute measurement of $P_{S/AS}$ was

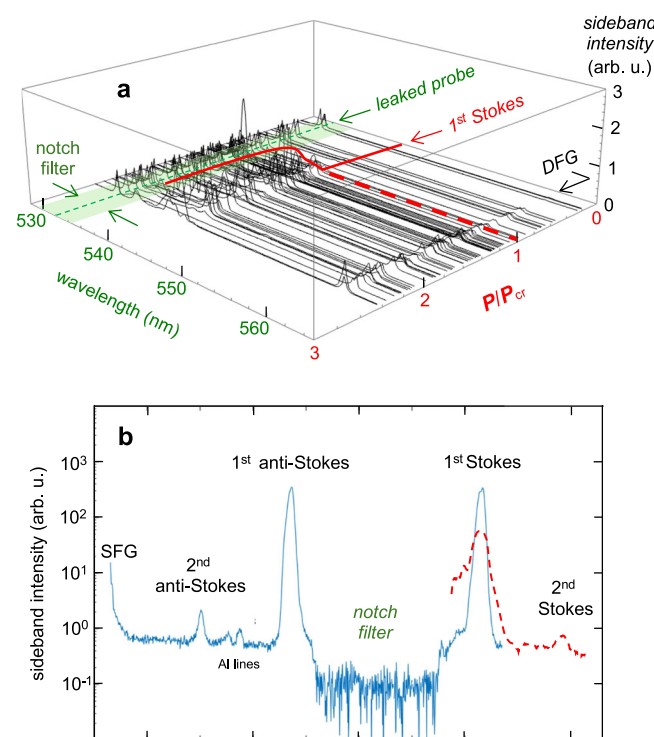

**b**

**Fig. 4 | Quantitative $P_L$-dependence of CTS. a** 1st Stokes intensity vs. $P_L/P_{cr}$ at $n_e = 1.3 \times 10^{18}$ cm$^{-3}$. Notch filter is rotated slightly to leak a small fraction of 532 nm probe light. 1st Stokes peaks at ~542 nm appear only for $P_L/P_{cr} > 1$; pump-probe difference-frequency-generation (DFG) peaks at ~560 nm are detectable down to $P_L \approx 0.5 P_{cr}$. **b** Examples of forward CTS spectra at same $n_e$ for two rare shots showing 2nd-order anti-Stokes (blue curve) or Stokes (dashed red curve) peaks along with corresponding 1st-order peaks. Weak spectral lines between 1st and 2nd anti-Stokes peaks originate from aluminum in gas nozzle that was vaporized by wings of the $CO_2$ pulse. Partial peak at ~505 nm is the pump-probe sum-frequency signal.

required, and the fraction of the probe that overlaps the wake was not accurately known. Moreover, $L$ was not well-characterized, and $F$ varied rapidly with $L$. However, on occasional shots for which second-order Stokes and/or anti-Stokes signals were visible − see e.g. Fig. 4b − a more accurate estimate was possible. Harmonics of $\delta n_e/n_e$, and thus of first-order S/AS sidebands, appear as $\delta n_e/n_e$ increases because the wave steepens[36–38]. Harmonic analysis for a cold plasma yields the ratio[37–39]

$$\delta n_e^{(1)}/n_e = \delta n_e^{(2)}/\delta n_e^{(1)}, \qquad (3)$$

where $\delta n_e^{(m)}$ ($m = 1, 2$) denotes the $m$th Fourier component of the wake oscillations and $n_e$ the uniform background plasma density. The amplitude $P_{S,AS}^{(2)}$ of the 2nd-order S/AS sideband is related to $\delta n_e^{(2)}$ by an equation of the form (2), with $P_{S,AS}^{(2)}, \delta n_e^{(2)}, \Delta k^{(2)}$ and $L^{(2)}$ replacing their first-order counterparts. In the approximations that $L^{(2)} \approx L$ and 1st- and 2nd-order wave vector matching factors average to similar values over multiple shots, the right-hand side of (3) can be approximated by $[P_{S,AS}^{(2)}/P_{S,AS}^{(1)}]^{1/2}$, i.e. the square-root of the 2nd-to-1st-order S/AS power ratio. The data in Fig. 4b, which is representative of a multi-shot average, shows $P_{S,AS}^{(2)}/P_{S,AS}^{(1)} \approx .01$, implying $\delta n_e^{(1)}/n_e \approx 0.1$ using Eq. (3). Since 2nd-order sidebands appeared on only ~5% of shots, we infer that $\delta n_e^{(1)}/n_e < 0.1$ for most shots driven by 0.5 TW pulses.

Figure 5a shows how the Stokes and anti-Stokes sideband profiles varied with pump-probe delay $\Delta t$ in increments of 0.33 ps at fixed $n_e = 1.5 \times 10^{18}$ cm$^{-3}$ and $P_L = 0.5$ TW. Both sidebands rise and fall within

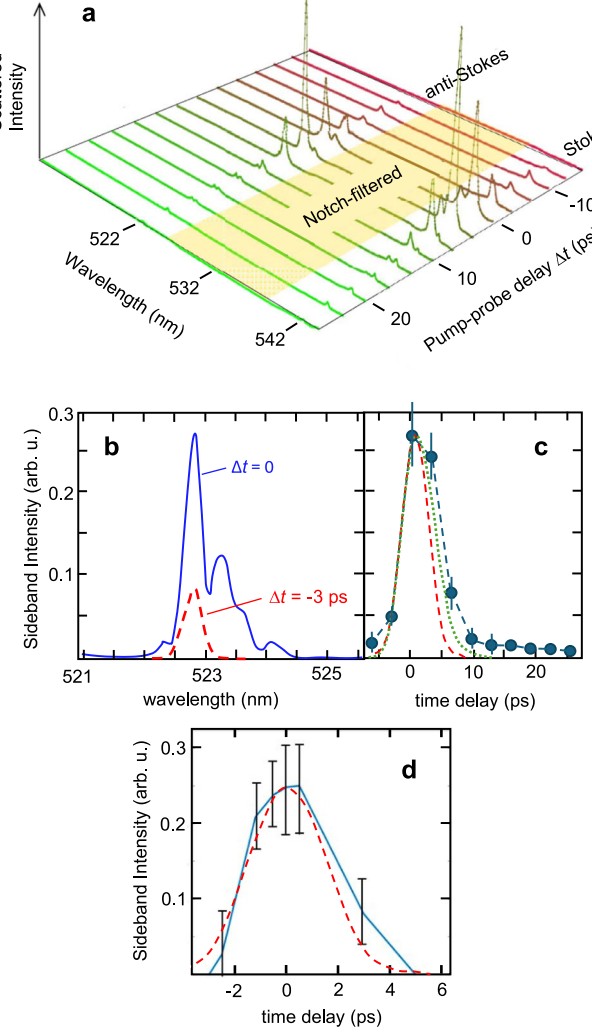

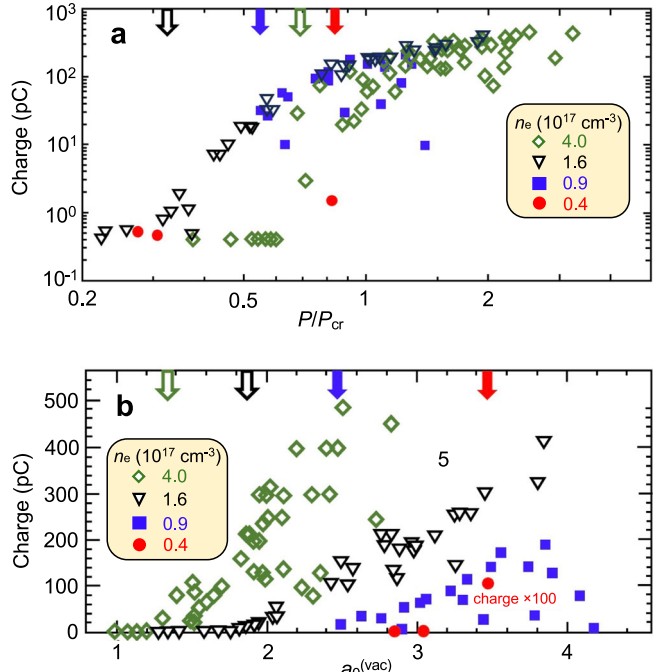

**Fig. 6 | Charge of accelerated electron bunches. a** Total charge of > 1 MeV electrons vs. normalized peak power $P_L/P_{cr}$ of ~ 2 ps laser pulses driving LWFA in plasma of 4 densities in the range $0.4 \times 10^{17} \leq n_e \leq 4 \times 10^{17}$ cm$^{-3}$ as per key. **b** Data from **a** replotted as a function of $a_0^{(vac)}$. Color-coded arrows at top of **a, b**: electron appearance thresholds for each $n_e$.

**Fig. 5 | Quantitative Δ*t*-dependence of CTS. a** 1st Stokes and anti-Stokes sideband intensity profiles vs. pump-probe delay $\Delta t$ at $n_e = 1.5 \times 10^{18}$ cm$^{-3}$ and $P_L = 0.5$ TW. **b** Example of spectrally-structured sideband at $\Delta t \approx 0$ (blue curve), in contrast to simpler single-peaked sideband observed at $|\Delta t| > 0$ (red dashed curve). **c** Blue data points: Plot of sideband intensity vs. $\Delta t$ for a different data run than **a**, but under identical conditions, averaged over first Stokes/anti-Stokes peaks and multiple shots. Blue dashed lines connect data points. Red dashed curve: pump-probe cross-correlation. Green dotted curve: simulated response from **d** convolved with pump-probe autocorrelation. **d** Simulated average Stokes/anti-Stokes sideband amplitude on continuous-wave probe ($\lambda_{pr} = 2.5\,\mu m$, field amplitude $E_{pr} = E_L/200$, focus $w_0^{pr} = 40\,\mu m$) co-propagating with a 2 J, 4 ps pump ($\lambda_L = 10.6\,\mu m$, $w_0 = 20\,\mu m$), with profile shown by red dashed curve, in $n_e = 10^{18}$ cm$^{-3}$ plasma at selected delays $\Delta t$. Error bars: 1 SD of variation among repeated experimental **c** or simulation **d** runs.

run from the CTS response at different locations within the probe with respect to the pump. The ~ 1 ps relaxation is consistent with the dephasing time of the inhomogeneously broadened Raman-shifted lines. The resulting plot (data points and blue connecting lines in Fig. 5d), however, is not convolved with the 4 ps probe used in the experiments. Including this probe convolution yields the dotted green curve in panel c). It is slightly broader than the pump-probe autocorrelation, shown by the red-dashed curve in panel c), but slightly narrower than the measured response. A weak component of both sidebands persisted out to $\Delta t \approx 30$ ps, suggesting a slower decay mechanism for low-amplitude wakes. The simulation could not capture this feature because of numerical noise. We have ruled out the possibility that post-pulses caused this delayed feature since autocorrelation measurements[12] do not detect any post-pulses in the interval $0 < \Delta t < 25$ ps, and those detected at longer $\Delta t$ are too weak to generate wakefields detectable by CTS. Its ~ 30 ps relaxation is, however, consistent with the decay time of electron plasma waves into ion acoustic waves[41].

## Measurements of accelerated electrons

Upon decreasing $CO_2$ laser pulse duration to $\tau_L \approx 2$ ps and increasing energy $\mathcal{E}_L$ to ≲ 4 J (i.e. $P_L \lesssim 2$ TW), with occasional pulses up to $P_L \sim 3$ TW, we observed collimated MeV electrons from plasmas of $n_e$ down to $3 \times 10^{16}$ cm$^{-3}$, 13 times lower than the threshold $n_e \approx 4 \times 10^{17}$ cm$^{-3}$ observed at lower $P_L$ by CTS. Electron yield peaked with the vacuum laser focus shifted forward toward the center, rather than exactly at the entrance, of the gas jet. We adjusted the exact vacuum focus location empirically with each run to maximize yield, but generally it lay between the entrance and center for experiments.

Figure 6 a plots total charge yield (determined from integrated luminescence from a calibrated[42] downstream scintillating screen) vs. $P_L/P_{cr}$ for ~ 100 shots of power $0.2 \lesssim P_L \lesssim 3.3$ TW driving plasmas of four different $n_e$, indicated in the legend. Approximately half of these

times close to the width of the pump-probe cross-correlation (see Fig. 5c). High-amplitude sidebands at $\Delta t \approx 0$ were spectrally multi-peaked (see Fig. 5b). Their strongest peaks occurred at the same frequency $\omega_{pr} \pm \omega_p$ as the single peak of lower-amplitude wakes (red curve, Fig. 5b), but satellite peaks at smaller Stokes/anti-Stokes shifts accompanied them. We attribute this spectral structure to transient $n_e$ gradients within the pump envelope due to its ponderomotive force on plasma electrons, which can lead to a distribution of Stokes/anti-Stokes shifts at $\Delta t \approx 0$. Cross-phase modulation of the probe by the strong pump may have also contributed[40]. Figure 5d shows results of a SPACE simulation that included a continuous-wave (CW) probe beam ($\lambda_{pr} = 2.5\,\mu m$) co-propagating with the 10 $\mu m$, 4 ps drive pulse, shown by the red dashed curve. Using a CW probe saved computational time, because the time response could be deduced in a single computational

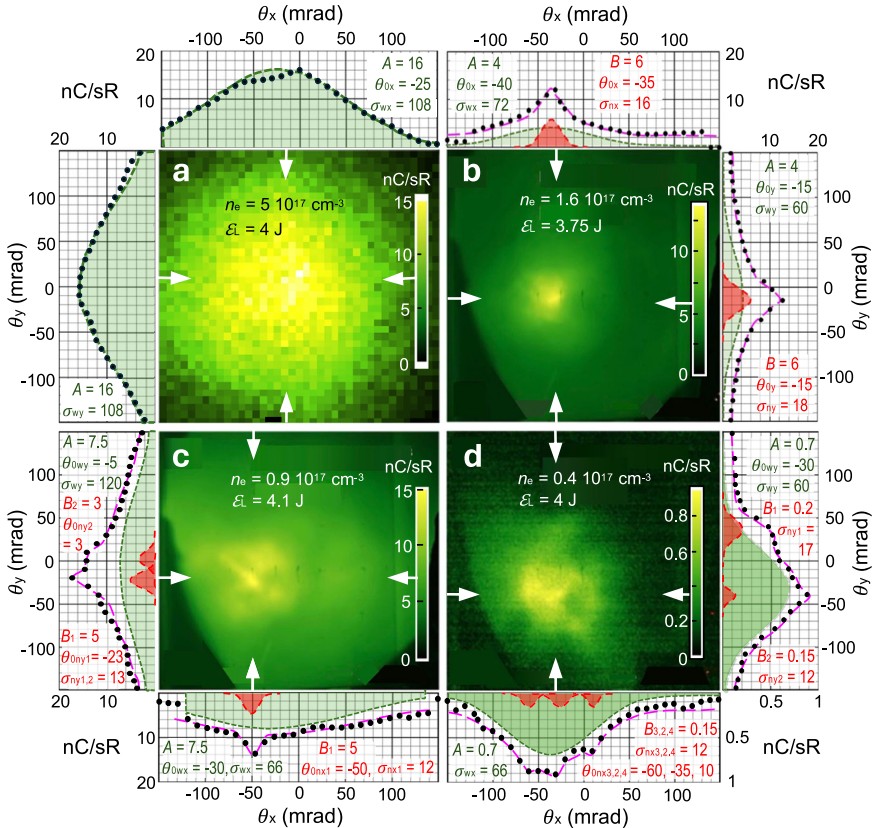

**Fig. 7 | Electron angular distributions. a** Simulated e-beam profile at $z = 27$ cm for conditions of Fig. 1d-f, and **b–d** luminescence images for the 3 lowest $n_e$ in Fig. 6 recorded on LANEX screen at $z = 27$ cm. White arrows: paths of line-outs displayed around periphery. Black dots: line-out data; long-dashed pink curves: fits of each lineout to sum of a wide ($w$) Gaussian $Ae^{-(\theta_i - \theta_{0i})^2/\sigma_{wi}^2}$ (green, $i = x, y$), one or more narrow ($n$) Gaussians $B_j e^{-(\theta_i - \theta_{0ij})^2/\sigma_{nij}^2}$ (red, $j$ = peak label), and background count $C$, assumed constant for each panel. Non-zero parameter values are listed inside each panel in units of nC/sR ($A, B_j$) or mrad ($\theta_{0i}, \sigma_{wi,nij}$). $C = 2$ nC/sR for **b** and **c** and 0.1 for **d**.

shots, shown by green diamonds, used pulses of power $0.2 \lesssim P_L \lesssim 1.5$ TW to drive $n_e \approx 4 \times 10^{17}$ cm⁻³ plasma, for which $P_{cr} = 0.5$ TW. At $P_L/P_{cr} < 0.7$ we observed only the detector noise floor at this $n_e$. But yield rose sharply for $P_L \gtrsim 0.7 P_{cr}$, and exceeded 100 pC for $P_L > P_{cr}$. Remaining data points show yield for shots up to 3.3 TW peak power driving less dense plasma: $n_e = 1.6 \times 10^{17}$ cm⁻³ ($P_{cr} = 1.5$ TW, inverted black triangles); $0.9 \times 10^{17}$ cm⁻³ ($P_{cr} = 2.6$ TW, filled blue squares; $0.4 \times 10^{17}$ cn⁻³ ($P_{cr} = 5.9$ TW, filled red circles). At all four $n_e$, the threshold power $P_L^{(thr)}$ for generating detectable charge, indicated by color-coded arrows along the upper horizontal axis of Fig. 6(a), was less than $P_{cr}$. At $n_e = 1.6 \times 10^{17}$ cm⁻³, $P_L^{(thr)}$ reaches as low as $0.3 P_{cr}$, more than $3 \times$ smaller than the wake formation threshold observed by CTS for longer, less powerful pulses in denser plasma (Fig. 1d–e). Such sub-$P_{cr}$ appearance thresholds indicate that self-focusing was no longer the principal mechanism triggering wake formation or acceleration at these low $n_e$.

To help understand these sub-$P_{cr}$ thresholds, Fig. 6b re-plots data in Fig. 6a as a function of vacuum focused $a_0^{(vac)}$ for each $n_e$. Now there are four $n_e$-dependent appearance thresholds at $a_0^{(vac)} = 1.35$ (for $n_e = 4 \times 10^{17}$ cm⁻³), 1.9 (for $n_e = 1.6 \times 10^{17}$ cm⁻³), 2.5 (for $n_e = 0.9 \times 10^{17}$ cm⁻³), and 3.4 (for $n_e = 0.4 \times 10^{17}$ cm⁻³). Moreover, all are relativistic. Evidently in this regime, $a_0$ is high enough to trigger wake formation without self-focusing. Indeed exponentially-growing self-modulated wakes driven by relativistically intense laser pulses with $P_L/P_{cr}$ as small as 0.25 have been seen in computer simulations (see e.g. Fig. 4 of[43]), but not, to our knowledge, previously realized in experiments. In contrast, under the conditions of e.g. Fig. 4a ($n_e = 1.3 \times 10^{18}$ cm⁻³, $P_{cr} = 0.15$ TW), $a_0^{(vac)} = 0.89$ at $P_L = P_{cr}$. Thus for $P_L < P_{cr}$, $a_0^{(vac)}$ was sub-relativistic, insufficient to drive growth of the forward Raman and

self-modulation instabilities rapidly enough to produce a detectable wake.

Figure 7a shows a simulated e-beam profile 27 cm downstream of the accelerator for the conditions of Fig. 1d–f ($n_e = 5 \times 10^{17}$ cm⁻³). The approximately Gaussian profile with $\Delta\theta_{HWHM} \approx 75$ mrad typified most observed profiles at this $n_e$. At lower $n_e$, on the other hand, about 90% of electron-producing shots revealed, in addition to a near-Gaussian background of similar HWHM, an intense core with divergence $40 \gtrsim \Delta\theta_{HWHM} \gtrsim 12$ mrad. Figure 7b–d shows examples of luminescence images recorded on a LANEX screen at $z = 27$ cm for $n_e = 1.6$, 0.9, and $0.4 \times 10^{17}$ cm⁻³. The narrowest such peaks had half-widths $\Delta\theta_{HWHM} \approx 12$ mrad. The centroids of these narrow beamlets exhibited RMS shot-to-shot pointing fluctuations of ~ 5 mrad.

To characterize the energy of electrons contributing to each part of this angular distribution, the intense core and a slice of the diffuse background passed through the 2 mm-wide vertical entrance slit of a magnetic spectrometer. The left-hand column of Fig. 8 shows energy- (horizontal) and angle- (vertical) resolved luminescence images recorded on LANEX screens at the spectrometer's detection plane, from electrons accelerated at $n_e = 2.4 \times 10^{17}$ cm⁻³ (a1) and $4.1 \times 10^{17}$ cm⁻³ (a2-a5). The right-hand column shows corresponding electron energy distribution curves, obtained by vertically integrating the raw data and re-scaling the horizontal axis to be linear in $E_e$. For the data in Fig. 8a, the LANEX screen was positioned diagonally across the spectrometer's dispersion plane, as shown by the green line in the inset of panel b1. Here, it recorded electrons of energies $0 < E_e < 7$ MeV. Beyond $E_e \sim 1$ MeV, electron yield decreased exponentially and angular spread narrowed with energy, mirroring thermal electron spectra typically observed from SM-LWFAs at higher $n_e$[17]. Similarly-shaped electron

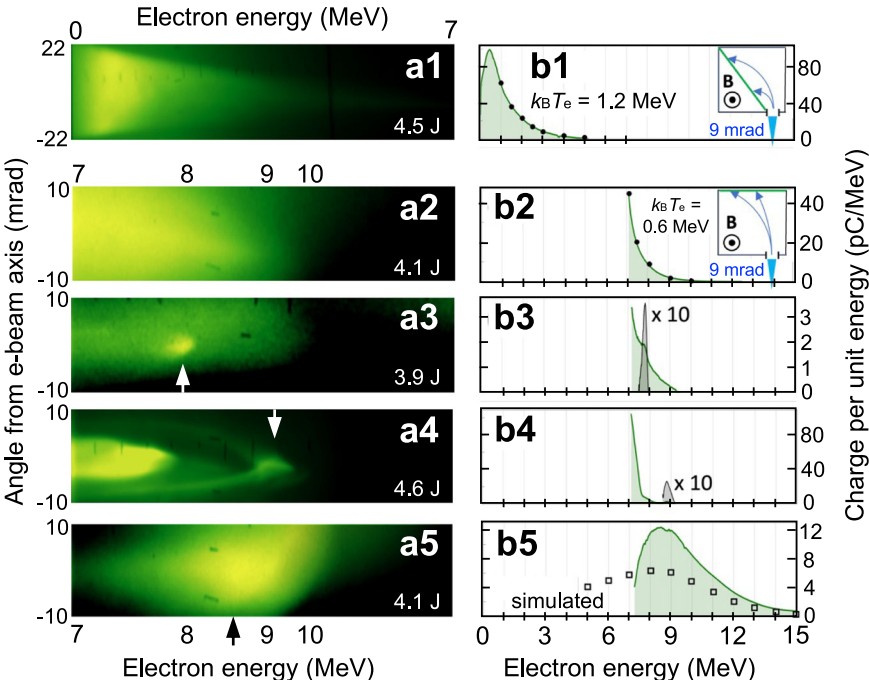

**Fig. 8 | Electron energy distributions.** LANEX luminescence images from magnetic spectrometer (left column) and corresponding energy distribution plots (right column). Row 1: Data taken with LANEX screen in position 1 (see inset of panel **b1**: solid green line) and $n_e = 2.4 \times 10^{17}$ cm$^{-3}$ to record electrons with energy $0 < E_e < 7$ MeV. Rows 2-5: Data taken with LANEX screen in position 2 (see inset of panel **b2**: solid green line) and $n_e = 4.1 \times 10^{17}$ cm$^{-3}$ to record electrons with energy $7 < E_e < 15$ MeV. In left column, horizontal $E_e$ scale at bottom applies only to Rows 2-5. Arrows in **a3** thru **a5** highlight quasi-monoenergetic features. Black dots in **b1** and **b2** are fits of the exponential tail of the energy distribution curve to functions $\exp(-E_e/k_B T_e)$, with $k_B T_e = 1.2(0.62)$ MeV for **b1** (**b2**). Open black squares in **b5** show the energy distribution of electrons from the simulation in Fig. 1d–f, for conditions ($n_e = 5 \times 10^{17}$ cm$^{-3}$, 4.0 J pump) close to those of the measured distribution in this panel.

spectra were observed in this energy-range for nearly all electron-yielding shots.

When the LANEX screen was re-positioned at the back of the spectrometer (green line at the top of the inset of panel b2), it recorded electrons of energies $7 < E_e < 15$ MeV, as rows 2-5 of Fig. 8 exemplify. Figure 8a2 displays an exponential distribution, possibly the high-energy tail of a distribution similar to the one shown in Fig. 7a1. Black dots in panels b1 and b2 show fits of the high-energy tail of these energy distributions to exponential functions $\exp(-E_e/k_B T_e)$, where $k_B$ is Boltzmann's constant, and $T_e$ is an effective electron temperature.

Remaining rows of Fig. 8 uniquely exhibit spectral peaks, which were observed on approximately 90 % of ~ 70 electron-yielding shots characterized with the spectrometer in its higher energy configuration. Supplementary Fig. 2 presents additional examples of electron spectra with both exponential and peaked spectra. Panels a3 and b3 present a sharp peak at $E_e \approx 7.7$ MeV (superposed on a broader background) containing ~ 0.1 pC charge with an angular half-width $\Delta\theta_{HWHM} \approx 2$ mrad in the vertical direction and a fractional horizontal width (HWHM) $\Delta E_e/E_e$ of only 0.03 (i.e. $E_e = 7.7 \pm 0.2$ MeV). Our observations of spectrally-integrated luminescence images such as Fig. 7b–d consistently show vertical and horizontal divergences of similar magnitude. Thus on the reasonable assumption that they are equal, this beamlet's inherent divergence is responsible for ~ 2/3 of the horizontal width of the recorded feature. When divergence is de-convolved, we obtain fractional energy spread $\Delta E_e/E_e \approx 0.01$ (i.e. $E_e = 7.7 \pm 0.07$ MeV), which rivals the smallest fractional energy spreads reported for bubble-regime LWFAs[44]. Panels a4 and b4 show a spectrum with two quasi-monoenergetic peaks, possibly due to electrons trapped in separate accelerating buckets of the wake. One is centered at $E_e \approx 7$ MeV with ~ 100 pC charge and $\Delta E_e/E_e \approx 0.15$, the other at $E_e \approx 8.5$ MeV with ~ 0.5 pC and $\Delta E_e/E_e \approx 0.01$ when de-convolved as described earlier. At the other extreme, in panels a5 and b5 we see a peak centered at

$E_e = 9$ MeV with $\theta \approx 7$ mrad containing 50 pC with $\Delta E_e/E_e \approx 0.35$. For comparison, the open black squares show the qualitatively similar peaked spectrum of electrons emerging from the simulated accelerator in Fig. 1d–f for similar conditions. This comparison shows that simulations reproduce peaked (as opposed to exponential) spectra observed on the high-energy screen with close to the observed peak energy, but do not capture the exceptionally narrow features shown in rows 3 and 4 of Fig. 8. Spectral peaks on other shots had energy/angular widths and charges in between these extremes. Quasi-monoenergetic peaks observed on the high-energy screen were narrower in angular spread than the exponential distributions observed on the low-energy screen, suggesting that they correspond to the bright cores of the images in Fig. 7b–d. The angular width of the quasi-monoenergetic peaks scaled approximately in proportion to its energy width.

## Discussion

The appearance of quasi-monoenergetic, collimated-electron spectral features on ~ 90% of electron-yielding shots suggests that for $n_e \lesssim 4 \times 10^{17}$ cm$^{-3}$ and $P_L > 0.5$ TW, we have entered a transitional regime between SM-LWFA and "forced"[28] or bubble-regime[21] LWFA in which the number $N_p = \omega_p \tau_L/2\pi$ of plasma periods within the drive pulse envelope is small, enabling drive pulse energy to channel preferentially into a single plasma period as it self-steepens. Here, $N_p$ ranges from ~ 10 for the conditions of rows 2 thru 5 of Fig. 8 ($n_e \approx 4 \times 10^{17}$ cm$^{-3}$) to ~ 3 for the conditions of Fig 7d ($n_e \approx 4 \times 10^{16}$ cm$^{-3}$). Although we have so far observed unambiguous quasi-monoenergetic spectral peaks only at the higher density (Fig. 8), where yield is higher, we observe the correlated intense beam cores down to the lower density (Fig. 7d). Importantly, here we observe signatures of this transitional regime at ~ 100 × lower $n_e$ than in its original discovery using $\lambda_L \approx 1\,\mu$m lasers[28] or in more recent experiments using $\lambda_L = 3.9\,\mu$m lasers[10]. Forced LWFA

at $\lambda_L \approx 1\,\mu m$ was a precursor of strongly nonlinear bubble-regime LWFA at the same $\lambda_L$, which for the first time provided plasma-based accelerating and focusing fields capable of preserving low $\Delta E_e/E_e$ and emittance, and which awaited only the development of shorter, more powerful $\lambda_L \approx 1\,\mu m$ drive pulse lasers a couple of years later. Similarly here, shorter ($\tau_L < 1$ ps), more powerful ($P_L \gtrsim 20$ TW) $\lambda \approx 10\,\mu m$ drive pulse laser technology capable of supporting bubble-regime LWFA at $n_e$ as low as $10^{16}$ cm$^{-3}$[24] now appears within reach (see Methods)[45,46]. The importance of achieving the bubble regime at $n_e \sim 10^{16}$ cm$^{-3}$ is two-fold: (1) high-quality, near-fully-blown-out bubbles can be generated with little or no uncontrolled background injection from the surrounding plasma[24], the principal origin of $> 1\%$ energy spreads and spin depolarization that currently limit applications of LWFA as e.g. free-electron-laser drivers or nuclear probes; (2) these bubbles are then large enough (e.g. bubble radius $R_b \approx \pi\lambda_p \approx 0.3$ mm at $n_e \approx 10^{16}$ cm$^{-3}$[47]) to accommodate precise, reproducible injection of sub-percent $\Delta E_e/E_e$, spin-polarized, high-charge electron bunches from a small conventional MeV linac, given current tolerances on pointing and timing jitter and bunch compression[48].

In summary, we have demonstrated SM and forced laser wakefield excitation and acceleration using a LWIR drive laser for the first time. Long-wave ($\lambda_L \approx 10\,\mu m$) excitation enables SM-LWFA at plasma densities $0.3 \times 10^{17} \lesssim n_e \lesssim 10^{18}$ cm$^{-3}$ that are $\gtrsim 100\times$ lower, and with wavelengths $\lambda_p$ that are $\sim 10\times$ larger, than previous SM-LWFA experiments using $\lambda_L \approx 1\,\mu m$ laser drive pulses. Although SM wakes are by definition excited non-resonantly (i.e. $c\tau_L >> \lambda_p/2$) in the longitudinal dimension, here for the first time we have excited them near-resonantly in the transverse dimension (i.e. $w_0 \approx \lambda_p/2$), generating stable SM wakes that can be accurately simulated. Accordingly our fully 3D PIC simulations have accurately modeled four key observables: (1) the SM-wake excitation threshold $P_L \approx P_{cr}$ for $n_e > 4 \times 10^{17}$ cm$^{-3}$ (see Fig. 1b–c); (2) the onset of self-injection for wakes excited by 2 TW pulses (see Fig. 1d–f); (3) the dependence of CTS sideband intensity (which is related to wake amplitude) on $n_e$ at fixed $P_L$ and $\Delta t$ (see Fig. 3b) and on (4) $\Delta t$ at fixed $P_L$ and $n_e$ (see Fig. 4a). This quantitative agreement, however, was obtained only when ionization and motion of hydrogen ions was taken into account. The quantitative agreement between measurements and simulations achieved here for SM-LWFA lends confidence to simulations[24] that predict efficient LWIR excitation of fully blown-out bubble regime wakes of unprecedented size ($\lambda_p \approx 300\,\mu m$) in $n_e \approx 10^{16}$ cm$^{-3}$ plasma − enabling unprecedented control over $e$-beam energy spread, emittance and polarization − if $CO_2$ lasers can be scaled to pulse durations $\tau_L < 1$ ps and peak powers $P_L > 20$ TW. Recent progress with CPA $CO_2$ laser technology suggests that this goal is within reach[45].

## Methods

### $CO_2$ laser

The terawatt $CO_2$ laser system at ATF is based on a master oscillator - power amplifier design with CPA[11,12]. An optical parametric amplifier (Quantronix Palitra) following a Ti:Sapphire oscillator- amplifier system generates 15 $\mu$J, 350 fs seed pulses. A grating stretcher chirps these to 140 ps and filters out a 0.8 THz wide part of their spectra sufficient for compression to 2 ps. Stretched pulses are transmitted to a 10 bar, mixed-isotope, discharge-pumped $CO_2$ regenerative amplifier (SDI Lasers, Ltd. HP-5) through a Faraday Isolator (Innovation Photonics FIO-5-9). After 16 double passes, a semiconductor switch, consisting of a Brewster-angle germanium plate activated by synchronized Nd:YAG laser pulses inside an optical cavity[49], couples out 10 mJ, 70 ps amplified pulses of high beam quality and transmits them through a 4-cm-thick NaCl window into an 8 bar final $CO_2$ amplifier (Optoel Co., PITER-I) that is also mixed-isotope and discharge-pumped. After propagating through a 5-meter, multi-pass gain length using nine intra-vessel mirrors, amplified pulses exit the power amplifier through a 10-cm-thick NaCl window with 10 cm clear aperture. An unfolded, mirror-less four-grating configuration with 100 lines/mm gratings and 70 % transmission temporally compresses them. Compressed 2 ps pulses of $> 5$ TW peak power were demonstrated. Peak power delivered to the interaction point is, however, presently limited to $\sim 2$ TW because of partially-in-air laser beam transport. Nonlinear optical interactions in these air-filled regions caused occasional pulses delivered with $P_L > 2$ TW to focus less tightly. This limitation will be eliminated in the near future after the installation of a vacuum compressor chamber and the completion of the vacuum transport line. See refs. 11,12 for further details.

The Supplementary Information presents measurements of the vacuum-focused spatial profiles, spectra, and duration of the $CO_2$ laser pulses. Supplementary Fig. 3 shows images of the vacuum-focused intensity profiles and their intensity-dependence. Supplementary Fig. 4 shows spectra of the $CO_2$ laser pulses in the 0.5 TW configuration used in initial wake generation experiments. These can be compared and contrasted with spectra in the 2 TW configuration, shown in Fig. 4 of ref. 12, used to accelerate electrons. Supplementary Fig. 5 shows how the duration $\tau_L$ of the $CO_2$ laser pulses, determined from single-shot autocorrelation measurements, varies with pulse energy.

Pulse energy was monitored by measuring the 8% reflection from a tilted uncoated NaCl input window of the vacuum system with a 95 mm diameter pyroelectric joule-meter (Gentec-EO, Model QE95LP). Laser pulse energy varied typically $\pm 20\%$ shot-to-shot. This natural variation combined with stepwise adjustment of the gain of the $CO_2$ laser amplifier (controlled by the voltage of the pumping discharge) was used for collecting pulse-energy statistics. The laser operates in a single-shot regime. The minimum time delay between shots, dictated by the capacitor charging time of the Marx generator powering the discharge in the final amplifier, is approximately 20 seconds. However, we deliberately extend this interval to at least 1.5 minutes between shots to allow adequate cooldown of the spark gaps (high voltage switching components). This practice helps to prolong their operational lifetime.

A research and development effort is presently underway aimed at $CO_2$ laser parameters needed for generating the blown-out bubble regime wakes ($\tau_L < 1$ ps, $P_L > 20$ TW). This goal can be achieved using a combination of two techniques[45]: 1) maximizing the bandwidth of the amplified pulses by simultaneous use of 9R and 9P branches of $CO_2$ gain spectrum, and 2) post-compression of the pulse. According to simulations backed up by recent proof-of-principle experiments[46] pulse durations in the order of 100 fs (3 optical cycles) with several joules of energy can be achieved with this approach in a system based on ATF's final amplifier. We are also in the initial stages of research and development aimed at increasing the laser repetition rate to 1 Hz and beyond. Currently, we are of the view that this increase would require a shift from electric-discharge to optical pumping methods[50].

### Probe laser

CTS probes originated as few mJ, 14 ps pulses of wavelength 1.06 $\mu m$ from a mode-locked Nd:YAG laser. After amplification in two passes to $\sim 30$ mJ, a polarizing beam-splitter divided them into orthogonally polarized sub-pulses, which passed through independent delay arms. They were re-combined with adjustable delay at a second polarizing beam splitter, and co-propagated into a potassium dihydrogen phosphate (KDP) crystal as ordinary ($o$) and extraordinary ($e$) pulses with different group velocities. Because of their group-velocity mismatch, the $o$ and $e$ pulses drifted through one another, during which time they underwent pump-depleted Type II second-harmonic generation. This effectively limited their overlap duration to 4 ps, creating a compressed probe pulse of wavelength $\lambda_{pr} = 0.532\,\mu m$, duration $\tau_{pr} = 4$ ps, and energy $\mathcal{E}_{pr} \approx 3$ mJ[51]. A lens focused these pulses with $f/62.5$ through a 5 mm-diameter hole in the pump OAP to a beam waist $w_0 \approx 50\,\mu m$ centered on the focal spot of the $CO_2$ pump pulse at the gas jet entrance.

Probe and $CO_2$ laser pulses were synchronized by phase-locking mode-locked pulse trains from the Nd:YAG and Ti:S oscillators at the

front ends of the probe and $CO_2$ laser systems, respectively, to a common frequency-divided RF reference frequency. Fast photodiodes monitored small portions of each oscillator output, delivered via piezoelectrically-controlled mirrors. These signals were sent to phase-locked-loops, which compared them to the RF reference signal and fed back error signals to the piezo-mirrors. This feedback adjusted the mirror positions to ensure phase locking of the pump and probe laser oscillators with the RF reference signal. The electronic synchronization loops at ATF are designed to operate with sub-picosecond timing jitter, a level that is negligible under the conditions of this work. The best evidence for this is that the rise time of pump-probe data in our experiments (e.g. Fig. 5c, blue data points) closely matched the pump-probe cross-correlation (dashed red curve) determined from single-shot autocorrelation measurements.

A delay arm within the $CO_2$ laser controlled pump-probe delay $\Delta t$ at the interaction region. We identified $\Delta t = 0$ using a silicon optical switch, i.e. we temporarily replaced the gas jet with a thin silicon plate, oriented so that probe and attenuated $CO_2$ pulses were near-normally incident. With the probe blocked, or trailing the $CO_2$ pulse, the latter passed through the plate, since its photon energy lies below the silicon band gap. When the probe led the $CO_2$ pulse, its above-gap photons generated a short-lived dense electron-hole plasma that reflected and absorbed the $CO_2$ pulse. A detector monitoring the transmitted $CO_2$ pulses as $\Delta t$ varied observed the sharp change in transmission at $\Delta t = 0$.

After the probe co-propagated through the gas jet with the 0.5 TW pump pulse, a BK7 vacuum chamber window blocked the pump. A lens collected transmitted green probe light and delivered it to a series combination of two notch interference filters (Alluxa, Inc.), each with optical density 6 within a 14 nm (FWHM) spectral window centered at $\lambda_{pr} = 0.532 \, \mu m$. Remaining probe light outside this window then entered an imaging spectrometer (SPEX model 270M) equipped with a charge-coupled device (CCD) camera (Princeton Instruments ProEM1024B) that detected a spectral region spanning 37 nm centered at 532 nm.

### Accelerated electron characterization

We characterized accelerated electrons generated with the vacuum laser focus between the entrance and the center of the gas jet, where we observed maximum electron yield. For energy-integrated measurements, the beam illuminated a scintillating screen (Kodak Lanex Regular) covered with a $20 \, \mu m$-thick aluminium laser shield located 27 cm downstream of the gas jet, close enough to capture the entire beam profile, but far enough to avoid saturation. Cathodoluminescence from the back of the screen was imaged to a CCD camera. We extracted total charge from integrated luminescence, using the calibration of[42].

For energy-resolved measurements, we constructed an in-vacuum magnetic electron spectrometer. Electrons entered the spectrometer through a 2 mm slit that limited their cone angle to 9 mrad. They passed through one of two cylindrical regions filled with uniform axial magnetic fields: 1) B = 0.25 T, diameter 5 cm, followed by a Lanex screen that allowed measurement of $> 5$ MeV electrons with 40 % resolution; 2) B = 0.3 T, diameter 10 cm, followed by a Lanex screen configured to measure either 0 - 7 MeV electrons with 10 % resolution or 7-15 MeV electrons with 10 % resolution at 10 MeV. The cited magnetic fields and their associated fringe fields were profiled with a Hall probe, and the measured fields used in calculating electron trajectories through the spectrometer. The 9 mrad cone angle of electrons entering the magnetic fields was, however, the main factor limiting energy resolution to 10%.

### Simulations

Simulations were performed using the 3D, parallel, relativistic PIC code SPACE[52]. The electromagnetic module of SPACE utilizes Yee's finite-difference time domain method for solving field equations[53] and the Boris-Vay pusher for advancing macroparticles[54,55]. SPACE also includes algorithms for atomic physics processes induced by high-energy laser- and beam-plasma interactions[27,52]. The algorithm for laser-induced tunneling ionization is based on ADK formalism[56]. The code computes ionization and recombination rates on the grid and transfers them to particles, rather than using a Monte Carlo approach. In contrast to simulations reported in ref. 27, which assumed stationary ions, the simulations in Fig. 1 (and ref. 24) take ion motion fully into account. This dampened wake amplitudes and raised self-injection thresholds compared to immobile-ion simulations. It also brought simulated injection thresholds into closer agreement with observed thresholds for relativistic electron production.

All simulations were performed in a 3D Cartesian geometry using a static window in the laboratory frame. The computational box has a transverse size of $600 \, \mu m$ and a longitudinal size of $3 - 5$ mm, with a transverse resolution of $dx = dy = 2.0 \, \mu m$ and a longitudinal resolution of $dz = 0.5 \, \mu m$. This corresponds to approximately 20 cells per laser wavelength ($\lambda_L$) in the longitudinal direction and 10 cells per beam waist $w_0$ in the transverse direction. Simulations use a minimum of 32 macro-particles per cell. Numerical convergence studies confirmed that this resolution is sufficient for the study of the problems targeted here.

Test simulations were carried out to optimize gas jet shape, laser pulse profile, and vacuum laser focus location used in presented simulation runs. These tests balanced four criteria: 1) best match to known experimental conditions, within measurement uncertainty; 2) best match to experimental results (sideband intensities, electron yield and spectrum); 3) best simulation efficiency; 4) least sensitivity to small changes in input parameters. Simulations of CTS experiments (Fig. 1b, c; Fig. 3b; Fig. 5d) came close to meeting all four criteria simultaneously. Simulations of electron acceleration experiments (Fig. 1d–f; Fig. 7a; Fig. 8b5) required some compromise. We made the following decisions:

- *Gas jet*: Simulation results were insensitive to entrance (exit) ramp length over the range $0 \leq L_{ent} \lesssim 0.4$ mm ($0.2 \leq L_{exit} \lesssim 0.4$ mm. Here, we present results with $L_{ent} = 0$ (in the interests of simulation efficiency) and $L_{exit} = 0.25$ mm (close to measured ramp). Overall jet length (including ramps) was fixed at the measured length of 2 mm.
- *Laser pulse profile*: A Gaussian vacuum spatial (temporal) profile with $w_0 = 27.5 \, \mu m$ ($\tau_L = 4$ or 2 ps) was used because it closely matched vacuum focus spot profile (autocorrelation) measurements (Supplementary Fig. 3c, inset; Supplementary Fig. 5 and ref. 12). These measurements provided no clear guidance for exploring deviations from Gaussian profiles.
- *Vacuum laser focus position*: In test simulations of CTS experiments, results were insensitive to small changes in vacuum focus location over the range $0 < z < 0.1$ mm (referring to the horizontal scale of Fig. 1a). In test simulations of electron acceleration experiments, for which the vacuum focus was shifted toward the gas jet center ($z = 1$ mm), results best matched experimental electron yield and energy for $z = 0.1$ mm. Thus, as a compromise, we used a vacuum focus at $z = 0.1$ mm for simulations of all experiments. This is smaller than the average $z$ used in electron acceleration experiments. We attribute the discrepancy tentatively to the highly nonlinear nature of the interaction in those experiments, as a result of which small deviations of gas jet and laser pulse profiles from their ideal shapes can play an outsized role in the results of the experiment.

### Data availability

Experimental data were generated at BNL's Accelerator Test Facility. The authors declare that all data supporting the findings of this study are available within the paper and its Supplementary Information. Additional inquiries about the data should be directed to the corresponding author.

## Code availability

The authors declare that the computer code SPACE supporting the findings of this study is fully documented within the paper, its references, and its Supplementary Information. Additional inquiries about the codes should be directed to R. S.

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

## Acknowledgements

This work was supported by grant DE-SC0014043 from the U.S. Department of Energy (DoE), Office of Science, High Energy Physics (V.N.L. and N.V.-N., principal investigators; M.C.D. and C.J., sub-awardees). U. Texas-Austin authors acknowledge additional support from Air Force Office of Scientific Research grant FA9550-16-1-0013 and DoE grant DE-SC0011617 (M.C.D., principal investigator). Brookhaven National Lab co-authors acknowledge additional support from DoE grant DE-SC0012704.

## Author contributions

R.Z. designed and installed the apparatus for optical wake probing and accelerated electron characterization, with assistance from J.W., Y.C., D.L.A., A.G., P.I., and I.P., and led the acquisition and analysis of all experimental data. N.V.-N. coordinated experimental activity with BNL staff and M.C.D., P.K. and A.C. carried out SPACE simulations under the supervision of R.S., who coordinated with R.Z. and M.C.D. on interpreting experimental results. D.L.A. and C.Z. contributed additional simulation support. M.B., M.F., R.K., K.K., M.N.P., and C.S. operated, maintained and upgraded the $CO_2$ laser system under the leadership of I.V.P. and M.A.P., M.C.D. conceived the experiments, inspired by theoretical work by I.V.P. on $CO_2$ laser-driven wakefield acceleration[13], led contributions of the U. Texas-Austin team and wrote the paper, in consultation with R.Z., R.S., C.J. and N.V.-N., V.N.L. acquired funding for the experiment, and contributed to discussions of procedure and analysis. C.J. contributed frequent advice and guidance based on years of experience with $CO_2$ laser experiments. All authors discussed the results and commented on the manuscript.

## Competing interests

The authors declare no competing interests.

## Additional information

**Peer review information** : *Nature Communications* thanks Hans-Peter Schlenvoigt and the other, anonymous, reviewers for their contribution to the peer review of this work. A peer review file is available.

