## [Peer Review File · Nature Communications]

Plasma electron acceleration driven by a long-wave-infrared laserReviewer #1 (Remarks to the Author):

Key results

The manuscript "Plasma electron acceleration driven by a long-wave-infrared laser" reports on experimental results, showing progress of laser-driven electron acceleration in underdense plasma by means of a CO₂ laser. Those laser types are, compared to widely established Ti:Sapphire lasers, of much less peak power. Therefore it was believed those systems, despite favorable scaling laws, could not yet achieve electron acceleration.

The manuscript shows first some estimates (by means of simulations) of laser guiding and plasma wave excitation up to the wave-breaking limit. This is followed by comprehensive plasma wave probing results by means of collective Thomson scattering, an established technique. The results are very consistent and well presented.

Thereafter follows the section of the main result, the characterization of the accelerated electrons. Firstly, a laser power scan is presented for different plasma densities, showing a clear unexpected signature for low plasma densities where a "too weak" initial laser power (compared to a density-dependent critical power) is sufficient to generate electrons. Secondly, electron spectra are presented, showing contributions of broad and narrow energy spreads. In fact, the spectrally narrow contributions appear to have extremely small energy spreads of per-cent level, while showing relatively high charges.

The paper main section closes with a discussion of the observations and a short, reasonable outlook.

Data, methodology and validity

Presented data, the methods to experimentally derive them, their analysis and supporting simulations are state-of-the art and well presented.

Clarity

The manuscript is of outstanding quality in terms of clarity. The historical excursus in the introduction is no divagation but sets very well scientific context. Also the retention of the main result of extremely small electron energy spread to quite the end is reasonable, since the wakefield characterization is prerequisite.

Yet a few minor suggestions remain to the authors.

- Page 2 column 2 lower half, where the transverse and longitudinal resonance conditions are discussed. It is written "even though the pulses were mismatched to the longitudinal resonant condition". It would help to quantify how much the mismatch is (in terms of a factor), as well how well the transverse condition was matched.
- Page 5 Fig 4 (a): there is a $\Delta \lambda_{pu-pr}$ shown but not further discussed, and not clear what the reference point of that difference is in the figure. The respective arrow seems to point to the band of DFG.
- Page 7 Fig 6: It might be more instructive to show both high and low density cases (a) and (b) in a single figure for better visual comparison. Vertical markers showing the respective thresholds might be used as well.
- The paragraph related to Fig 6 c "To help understand..." is not explaining well since the difference between 6a and 6b is not discussed but only the data in 6b. First, the splitting of the single data group in Fig 6b into 3 groups in Fig 6c is explained by Eq 1. If data is homogeneous in a plot against P/P_{cr} and there is a density-dependent P_{cr} , whereas P is only bound to a_0 (both are vacuum observables, before possible modulations), data MUST split into density-dependent groups. Hence that does not explain to me the lowered threshold in units P/P_{cr} for the lower-density group. The authors write that a_0 is relativistic for the case of Fig 6b and 6c, yet a comparison to Fig 6a is missing. If a_0 is also relativistic, the argument is obsolete. Also the current sentences "the weak self-focusing ... undoubtedly raises a_0 " and "... a_0 is high enough ... without ... self-focusing" are contradicting. My feeling is that the key is here the difference in threshold powers between high and low density cases. For high density,

a_0 is smaller and thus the a_0 at the threshold is smaller than for lower-density shots. Yet, as the authors argue, with higher density is stronger self-focusing, and it cannot be excluded that again relativistic a_0 can be obtained for the high-density case. Therefore the argument might be that already only the vacuum parameters are such that for lower densities (6b) wake formation can be triggered whereas for high density (6a) the vacuum strength alone is insufficient. Yet, the value of a_0 needs to be cited again.

- Page 8, Discussion: The section starts with "The frequent appearance..." where a value for the relative frequency would support this statement.
- Page 8, right column, end of first paragraph "... tolerances on pointing jitter and bunch compression." I think that references to those values should be provided for transparency.

Significance

The results shown are of high significance for laser-plasma electron acceleration. Due to long wavelengths and low densities, large "bubbles" (the accelerating structures) can be generated which allow for de-coupling of electron injection and acceleration by easing the accuracy requirements at the accelerating stage. In particular, low energy spreads are of high importance for applications. The chosen technology platform is currently not widely established for immediate application experiments or reproduction of the results. Yet the technology is available and can be implemented where needed. I do not see this as a showstopper for publishing the results.

Reviewer #2 (Remarks to the Author):

Reviewer: Alexander S. Pirozhkov

The manuscript describes experimental results, simulations, and analysis of wakewave excitation and electron acceleration experiments driven by ~ 10 - μm -wavelength, high-power (up to 3 TW), short-pulse (down to 2 ps), CO₂ laser. The results extend notably the driving laser wavelength used in such experiments till now: 0.8 μm wavelength used routinely, 3.9 μm wavelength laser used in Ref. [10]. With longer wavelength, the acceleration at rather low plasma densities ($\sim 10^{17} \text{ cm}^{-3}$) can be driven by relatively low-power lasers (a few TW). Low densities open perspectives for potentially sub-mm accelerating structures: compared to several to tens-of- μm structures used typically, sub-mm scale will allow easier coupling with linear accelerators and easier diagnostics. Importantly, driving laser in this work is a discharge-pumped gas laser, which is very different from the used-up-to-now solid-state lasers. This is a long-expected milestone for the high-power CPA CO₂ lasers, which opens several potential applications including those apart from compact electron accelerators.

In addition to the key results summarized above, the authors explained several experimental observations and clarified the physics of laser-plasma interaction under the experimental conditions, namely excitation of the wakewave in the self-modulated (long-pulse: 4 ps) regime and estimated its threshold conditions and wave amplitude using sidebands of collective Thomson scattering; estimated the self-injection threshold; explained the role of relativistic self-focusing in the electron acceleration in the short-pulse (2 ps) regime. The authors analysed the electron spectra in the range of sub-MeV to 15 MeV: the spectra were mostly exponential but often contained quasi-monoenergetic features with a few percent relative bandwidth, and electron beam profiles, which exhibited divergent halo (typically 100 mrad) with presence of much lower-divergence, much brighter beamlets near the centre. The authors argued that these features corresponded to the onset of the forced wakefield regime, and that the strongly nonlinear bubble regime can be reached with a not-so-distant upgrade to ~ 20 TW, < 1 ps, which would provide improvements in the electron beam quality.

The present version of the manuscript gives a clear overall understanding of what was done, however, there are several unclear places and figures, which I enumerate below. Also, the relevance of the simulations should be emphasized by comparing the initial setup and results with the experiment on

the same graphs. There are also a few optional comments driven by my curiosity, however, I expect readers will have similar desires to know. Thus, why not to add a few (possibly supplementary) figures?

1. Page 1: "the momentum ratio $a_0 = e E_L / w_L m_e c$ ". The a_0 quantity is usually called dimensionless amplitude, why not to use this standard term?
2. Page 2: "an off-axis parabola (OAP) mirror focused linearly-polarized drive pulses from ATF's CPA CO2 laser [11, 12] to spot size $w_0 \sim 27.5 \mu\text{m}$ at the entrance of a supersonic hydrogen gas jet". Please clarify more precisely how far away from the gas jet centre the focal plane was located, or, even better, show it in Fig. 1(a); am I correct in my assumption that the focal plane in the simulation was located at the same position?
3. Page 2, please correct the inline formula " $\lambda_p/2 = k_p/\pi$ " (should be π/k_p).
4. Page 3: "Simulations of the interaction that include tunneling ionization, exemplified by Fig. 1(b)-(d), show that under these conditions, the wings of the focused CO2 laser pulse self-ionize a hydrogen column of radius $R_p \sim 50 \mu\text{m}$ ": it is unclear if this is relevant to the experiment, because the ionization radius depends on the spot shape at low-intensity (far-from-the-peak) regions. For a Gaussian spot, the ionization radius would be relatively small, for a Lorentzian spot of the same Full Width at Half Maximum (FWHM), it would be larger, for a non-ideal spot due to aberrations, spatiotemporal couplings, etc., the ionization radius can be many times larger than for an ideal spot. (Please also see my related comments #19 and #24).
5. The authors discuss the self-focusing threshold up to the middle of the first paragraph on page 3, and then abruptly use the self-injection threshold at the end of this paragraph and in Fig. 1. Inattentive readers like me will miss this difference and, like me, will be greatly confused. Thus, 0.5 TW is above self-focusing threshold for $n_e = 5 \times 10^{17} \text{ cm}^{-3}$, but below the self-injection threshold. Please explain and possibly discuss this for the benefit of the readers.
6. Page 3: "Our simulations showed a strong influence of the dynamic ionization model on the structure and evolution of the SM wakes compared to the pre-ionized plasma approximation [27]. Likewise, including the ion motion in simulation leads to the formation of ionization channels (Fig. 1 and [24]) which are largely suppressed in simulations using the fixed ion approximation [27]." These statements represent results that are not the most important for this work but still of interest. Unfortunately, they are not supported by figures. It would be nice to add comparison with pre-ionized and stationary-ion cases, probably as supplementary figures.
7. Fig. 1(c,e), the colour scales showing the electron momenta are not clear. The electric field is possible to see, although even this could be improved (for example, blue is a part of the momentum colour scale, as well as colour of the field curve). It might be better not to overlap the momenta and fields, although the authors may use other methods to clarify the figure.
8. As far as I understand, Fig. 1(d,e) shows simulations with conditions close to the experiment with 2 TW pulses where the electrons were accelerated, Figs. 6 and 7. Please compare simulated and experimental electron spectra and divergences/beam profiles.
9. Page 4: "Through such measurements we confirmed, as shown in Fig. 3(b), that sideband intensity within this window, was vanishingly weak or absent for $n_e < 4 \times 10^{17} \text{ cm}^{-3}$ for excitation at P_L . From Eq. 1, this "turn-on" density corresponds to $P_{cr} = P_L = 0.5 \text{ TW}$." However, the laser intensity is relativistic even without self-focusing, $a_0 \sim 2.2$, as written at the bottom of page 2. Thus, I would expect some wakewave excitation anyway, accompanied by corresponding measurable sidebands. What is wrong in my logic?

10. Fig. 2 caption: " n_e -independent sum- and difference-frequency peaks". However, it is known that the laser pulse frequency gradually downshifts due to energy transfer to plasma waves, at least in the bubble regime [Nie et al., Nature Communications 11 2787 (2020)], and this downshift is density-dependent. Can the authors check if this effect can be observed, and describe its absence (not measurable within ???nm detection threshold) or presence (above ???nm detection threshold)?

11. Fig. 4 caption: "pump-probe difference-frequency peaks at ~ 560 nm appear at all P_L ." This contradicts text on page 5: "DFG signals were observed at normalized power down to $P_L/P_{cr} \sim 0.5$."

12. Fig. 5: the simulated curve shown in (d) should be overlapped with the experimental curve shown in (c), and the apparent difference in their widths explained or at least discussed. Also, the pump-probe cross-correlation curve in Fig. 5(c) can be scaled to have its peak similar to the data points; that would reduce apparent discrepancy in temporal width.

13. Fig. 6(b): it would be nice to colour the data points according to the density, similar to (c).

14. Page 7: "The threshold power P_L for generating detectable charge was now only $0.3P_{cr}$, more than 3x smaller than the wake formation threshold observed by CTS for longer, less powerful pulses [Fig. 1(d)-(e)]. Such a low appearance threshold indicates that self-focusing was no longer the principal mechanism triggering wake formation or acceleration. To help understand this threshold, Fig. 6(c) re-plots data in Fig. 6(b) as a function of vacuum focused a_0 for three fixed n_e . Now there are three n_e -dependent appearance thresholds at $a_0(vac) = 1.9$ (for $n_e = 1.6e17 \text{ cm}^{-3}$), 2.4 (for $n_e = 0.9e17 \text{ cm}^{-3}$), and 3.4 for $n_e = 0.4e17 \text{ cm}^{-3}$." However, looking at the single trend on Fig. 6(b), and three different trends (different for each n_e) on Fig. 6(c), I would guess that P/P_{crit} ratio is more important than a_0 , meaning that the self-focusing is still the dominant governing process. Although, the authors are correct in emphasising that the value $P/P_{crit} \sim 0.3$, rather than ~ 1 , is a bit surprising.

15. Page 7: "an intense core with a divergence as small as ~ 10 mrad." However, I do not see 10 mrad divergence in Figs. 6(d-f); it may be due to the colour scale used. It would be nice to show (linear-scale) lineouts on top of the images - that would help to reveal the small-divergence core the authors write about, and discriminate it from the ~ 100 mrad diffuse background.

16. Page 7: "Fig. 7(a2) displays an exponential distribution, possibly the high-energy tail of a distribution similar to the one shown in Fig. 7(a1)." It would be of interest to know the temperatures of the exponential electron spectra.

17. Page 8: "When de-convolved from the spectrometer's instrumental width due to its 2mm entrance slit, this corresponds to fractional energy spread $dE_e/E_e \sim 0.01$." However, it seems that this electron beamlet divergence is small enough to pass thru the slit without clipping, and thus the slit itself should not affect the recorded energy width (the slit affects the absolute energy error bar, actually - please show it, by the way). On the other hand, the electron beamlet divergence in the horizontal direction does affect the apparent energy width. A deconvolution can be performed assuming the same horizontal and vertical divergence, although I am not sure if such assumption can be justified in this particular case: a supporting fact in favour of this hypothesis is that "The angular width of the quasi-monoenergetic peaks scaled approximately in proportion to its energy width," mentioned below. The same comment applies to the data shown in Fig. 7(a4,b4).

18. Page 8 Summary: "Accordingly our fully 3D PIC simulations have accurately modeled four key observables: (1) the SM-wake excitation threshold $P_L \sim P_{cr}$; (2) the self-injection threshold $a_0 \sim 3$; (3) the dependence of wake amplitude on n_e at fixed P_L and dt , and on (d) dt at fixed P_L and n_e ." However, (1) requires some explanation, see my point #5 above; (2) this threshold is not discussed in the text and is not obvious from the figures; (3) it is not obvious for me where this dependence is shown: is it Fig. 3(b)? but in this case, this is dependence of the CTS intensity vs.

density; of course, CTS intensity is connected with the wake amplitude, as described in the text, but they are not identical; (d) - probably, this should read (4) - again, I do not see the dependence of the wake amplitude on dt , only the normalized sideband amplitude on dt in Fig. 5(d).

19. Methods: CO2 laser: the measured focal spot, pulse shape, and spectrum, and their typical shot-to-shot and day-to-day variations should be shown, for ~ 0.5 TW and ~ 2 TW laser modes. Also, in the text it was mentioned that the spot quality degraded at higher laser powers: both "good" and "degraded" spots should be shown. This is necessary because propagation and self-focusing of a laser pulse in low-density plasma is strongly affected by the spot and pulse shapes. Without these figures and discussion, the results cannot be reproduced.

20. Methods: CO2 laser: please mention the repetition rate used in the experiment. Is it determined by the laser or experimental setup?

21. Methods: Probe laser: "These signals were sent to phase-locked-loops, which compared them to the RF reference signal and fed back error signals to the piezo-mirrors, which in turn adjusted the mirror positions to maintain fixed relative phases." As far as I understand, the intended meaning of this sentence is that the pump and probe laser oscillators were both synchronised to the same RF wave, and the "fixed relative phases" are relative to this wave. However, it is ambiguous, because my first impression that the pump and probe lasers were synchronised such that their relative phases were fixed, implying sub-cycle accuracy. I suggest to re-word the description to avoid misunderstanding.

22. Methods: Probe laser: it would be nice to show a figure showing the scan determining the delay-zero timing; if it was published somewhere, a reference would do. What is the delay-zero timing accuracy?

23. Methods: Accelerated electron characterization: please describe if the actual 3D magnetic field distributions were measured, or nominal fields and sizes were used in electron spectrometer energy calibration.

24. Methods: Simulations: Please add plasma-free (vacuum) spot and pulse shapes used in simulations to the figures showing measured spot and pulse, for comparison. The self-focused spot shapes would also be of interest.

25. Methods or main text: please describe how the gas jet density profile was obtained. The present description on page 4, "calibrated by independent optical measurements of the ionized gas jet," is too brief and does not reveal the method. What is the density error bar?

Reviewer #3 (Remarks to the Author):

This is an important work, it shows electron trapping and acceleration with a long wavelength laser driver in a plasma accelerator for the first time. This was achieved by having a high peak power laser as well as close to transverse matching to the plasma. Simulations of the high peak power CO2 laser pulses demonstrate that wavebreaking is expected for the upgraded laser case only however, a plasma wave is excited in both cases via the self-modulation instability. Detailed measurements of a laser probing the interaction which scattered off the plasma presented which allowed for an estimate of the plasma wave amplitude. Finally, the accelerated electron beam was characterised with an investigation into the conditions that trigger injection in the transition between self-modulated and blow out regimes.

A large volume of data was collected over a broad range of parameter space and its analysis is well communicated to the reader. This allows for a high degree of confidence in the trends and conclusions. The paper is well written and method of analysis sound.

Below follow comments and suggestions for further improvement.

- Abstract gives neutral gas backing pressures of target in atmospheres however, the remainder of the paper discusses plasma electron number density. Since demonstrating wave excitation at low density is a key outcome of the paper, I think that it would be easier to read if the abstract was consistent with the remainder of the paper.
- The calculation of normalised laser intensity should be checked. I could not reproduce $a_0 \approx 2.2$ for 2J, 4 ps, $\lambda = 10.3 \mu\text{m}$ and $\omega_0 = 27.5 \mu\text{m}$. Instead, I get $a_0 \approx 1.8$.
- For the laser intensity with the upgraded laser, it should be made clearer that this is the peak intensity assuming the highest overserved laser pulse energy, not the typical pulse energy. In this case, I can reproduce the intensity given in the text.
- In the text and figure 1, it is shown that electron bunches are accelerated by the 2 TW laser but no other information is given other than the maximum electron energy. What is the energy distribution of these electrons? How does it compare to the experimental measurements?
- In figures 3(b) and (c) the sideband intensity could be normalised to better compare where the peaks occur between the time delay and Stokes peaks to cross section respectively.
- Figure 4(a) is difficult to interpret, particularly the trend in red.
- An explanation should be given for the multiple injection events from the interaction. Do the simulations give an insight?
- A statement should be given as to whether the simulations also seen the narrow energy spread beam within the overall accelerated electron population.
- A comment should be made on the pointing stability of the electron beam. How many shots containing a narrow divergence beam are expected to overlap with the position of the entrance slit of the energy spectrometer magnet?
- It is stated that the pulse duration varied "significantly" from shot-to-shot, was the pulse duration measured on shot to correctly determine the peak power of that shot?
- In the simulations, what was the position of the (vacuum) laser focus? Experimentally, the laser was focused at both the beginning and centre of the gas jet.
Were the simulations performed with a static simulation window?

Key to type: 11 pt. *italic type: reviewer comments*

11 pt. standard type: our responses to Rev. 1 (green), Rev. 2 (red), Rev. 3 (blue)

Boldface type: description of manuscript changes in response to comments

Reference numbers in responses below are those of the revised manuscript.

Response to Reviewer #1.

Reviewer #1 comment 1: *Key results. The manuscript “Plasma electron acceleration driven by a long-wave-infrared laser” reports on experimental results, showing progress of laser-driven electron acceleration in underdense plasma by means of a CO2 laser. Those laser types are, compared to widely established Ti:Sapphire lasers, of much less peak power. Therefore it was believed those systems, despite favorable scaling laws, could not yet achieve electron acceleration.*

The manuscript shows first some estimates (by means of simulations) of laser guiding and plasma wave excitation up to the wave-breaking limit. This is followed by comprehensive plasma wave probing results by means of collective Thomson scattering, an established technique. The results are very consistent and well presented.

Thereafter follows the section of the main result, the characterization of the accelerated electrons. Firstly, a laser power scan is presented for different plasma densities, showing a clear unexpected signature for low plasma densities where a “too weak” initial laser power (compared to a density-dependent critical power) is sufficient to generate electrons. Secondly, electron spectra are presented, showing contributions of broad and narrow energy spreads. In fact, the spectrally narrow contributions appear to have extremely small energy spreads of per-cent level, while showing relatively high charges.

The paper main section closes with a discussion of the observations and a short, reasonable outlook.

Our response: We appreciate this summary, which correctly identifies the key results.

Reviewer #1 comment 2: *Data, methodology and validity. Presented data, the methods to experimentally derive them, their analysis and supporting simulations are state-of-the art and well presented.*

Our response: We appreciate this positive assessment.

Reviewer #1 comment 3: *Clarity. The manuscript is of outstanding quality in terms of clarity. The historical excursus in the introduction is no divagation but sets very well scientific context. Also the retention of the main result of extremely small electron energy spread to quite the end is reasonable, since the wakefield characterization is prerequisite.*

Our response: We appreciate this positive assessment of the manuscript's clarity and organization.

Reviewer #1 comment 4: *Yet a few minor suggestions remain to the authors. • Page 2 column 2 lower half, where the transverse and longitudinal resonance conditions are discussed. It is written “even though the pulses were mismatched to the longitudinal resonant condition”. It would help to quantify how much the mismatch is (in terms of a factor), as well how well the transverse condition was matched.*

Our response: We added the phrases in bold-faced type: "...and thus satisfied a *transverse* near-resonant condition $k_p w_0 \sim \pi$ to within a factor of two over the density range $10^{17} \lesssim n_e \lesssim 10^{18} \text{ cm}^{-3}$ of interest here, even though the pulses were mismatched to the *longitudinal* resonant condition by factors ranging from 7 (for $n_e = 4 \times 10^{16} \text{ cm}^{-3}$, $\tau_L = 2 \text{ ps}$) to 100 (for $n_e = 2 \times 10^{18} \text{ cm}^{-3}$, $\tau_L = 4 \text{ ps}$).

Reviewer #1 comment 5: • Page 5 Fig 4 (a): there is a Delta lambda_pu-pr shown but not further discussed, and not clear what the reference point of that difference is in the figure. The respective arrow seems to point to the band of DFG.

Our response: Thanks for pointing out this unclear label. **We changed the label to "DFG", and added a definition of this acronym to the text and the captions of Figs. 2 and 4.**

Reviewer #1 comment 6: • Page 7 Fig 6: It might be more instructive to show both high and low density cases (a) and (b) in a single figure for better visual comparison. Vertical markers showing the respective thresholds might be used as well.

Our response: **We have taken the reviewer's suggestion by combining panels (a) and (b) of Fig. 6, using different colors to distinguish high from low density data points, and adding vertical markers to show thresholds. We correspondingly revised the text to discuss the data in these two panels together, which helps to address the reviewer's next comment. The revised text now reads:**

Fig. 6(a) plots total charge yield (determined from integrated luminescence from a calibrated [49] downstream scintillating screen) vs. P_L/P_{cr} for ~ 100 shots of power $0.2 \lesssim P_L \lesssim 3.3$ TW driving plasmas of different n_e . **Approximately half of these shots, shown by green diamonds, used pulses of power $0.2 \lesssim P_L \lesssim 1.5$ TW to drive $n_e \approx 4 \times 10^{17} \text{ cm}^{-3}$ plasma, for which $P_{cr} = 0.5$ TW. At $P_L/P_{cr} < 0.7$ we observed only the detector noise floor at this n_e . But yield rose sharply for $P_L \gtrsim 0.7P_{cr}$, and exceeded 100 pC for $P_L > P_{cr}$. Remaining data points show yield for shots of higher power ($2 \leq P_L \leq 3.3$ TW) driving less dense plasma: $n_e = 1.6 \times 10^{17} \text{ cm}^{-3}$ ($P_{cr} = 1.5$ TW, inverted black triangles); $0.9 \times 10^{17} \text{ cm}^{-3}$ ($P_{cr} = 2.6$ TW, filled blue squares); $0.4 \times 10^{17} \text{ cm}^{-3}$ ($P_{cr} = 5.9$ TW, filled red circles). At all four n_e the threshold power $P_L^{(thr)}$ for generating detectable charge, indicated by color-coded vertical arrows along the upper horizontal axis of Fig. 6(a), was less than P_{cr} . At $n_e = 1.6 \times 10^{17} \text{ cm}^{-3}$, $P_L^{(thr)}$ reaches as low as $0.3P_{cr}$, more than $3\times$ smaller than the wake formation threshold observed by CTS for longer, less powerful pulses in denser plasma [Fig. 1(d)-(e)]. Such sub- P_{cr} appearance thresholds indicate that self-focusing was no longer the principal mechanism triggering wake formation or electron injection and acceleration at these low n_e .**

Reviewer #1 comment 7: • The paragraph related to Fig 6 c “To help understand...” is not explaining well since the difference between 6a and 6b is not discussed but only the data in 6b. First, the splitting of the single data group in Fig 6b into 3 groups in Fig 6c is explained by Eq 1. If data is homogeneous in a plot against P/P_{cr} and there is a density-dependent P_{cr} , whereas P is only bound to a_0 (both are vacuum observables, before possible modulations), data MUST split into density-dependent groups. Hence that does not explain to me the lowered threshold in units P/P_{cr} for the lower-density group. The authors write that a_0 is relativistic for the case of Fig 6b and 6c, yet a comparison to Fig 6a is missing. If a_0 is also relativistic, the argument is obsolete. Also the current sentences “the weak self-focusing ... undoubtedly raises a_0 ” and “... a_0 is high enough ... without ... self-focusing” are contradicting. My feeling is that the key is here the difference in threshold powers between high and low density cases. For high density, P_{cr} is smaller and thus the a_0 at the threshold is smaller than for lower-density shots. Yet, as the authors argue, with higher density is stronger self-focusing, and it cannot be excluded that again relativistic a_0 can be obtained for the high-density case. Therefore the argument might be that already only the vacuum parameters are such that for lower densities (6b) wake formation can be triggered whereas for high density (6a) the vacuum strength alone is insufficient. Yet, the value of a_0 needs to be cited again.

Our response: We agree with the reviewer that the artificial distinction of the former Figs. 6a and 6b caused unnecessary confusion. **We have revised the paragraph in question to discuss data in the former Figs. 6a and 6b (now combined into a unified Fig. 6a) together. The new Fig. 6b (formerly 6c), plotting charge vs. a_0 , also includes data for all four n_e . The new discussion cites a_0 values throughout, and the contradiction that the reviewer notes above is removed. The revised paragraph reads:**

To help understand these sub- P_{cr} thresholds, Fig. 6(b) re-plots data in Fig. 6(a) as a function of vacuum focused $a_0^{(vac)}$ for each n_e . Now there are four n_e -dependent appearance thresholds at $a_0^{(vac)} = 1.4$ (for $n_e = 4 \times 10^{17} \text{ cm}^{-3}$), 1.9 (for $n_e = 1.6 \times 10^{17} \text{ cm}^{-3}$), 2.5 (for $n_e = 0.9 \times 10^{17} \text{ cm}^{-3}$), and 3.4 (for $n_e = 0.4 \times 10^{17} \text{ cm}^{-3}$). Moreover, all are relativistic. Evidently in this regime, a_0 is high enough to trigger wake formation and inject electrons without self-focusing. **Indeed exponentially-growing self-modulated wakes driven by relativistically intense laser pulses with P_L/P_{cr} as small as 0.25 have been seen in computer simulations (see e.g. Fig. 4 of [50]), but not, to our knowledge, previously realized in experiments. In contrast,**

under the conditions of e.g. Fig. 4(a) ($n_e = 1.3 \times 10^{18} \text{ cm}^{-3}$, $P_{\text{cr}} = 0.15 \text{ TW}$), $\alpha_0^{(\text{vac})} = 0.89$ at $P_L = P_{\text{cr}}$. Thus for $P_L < P_{\text{cr}}$, $\alpha_0^{(\text{vac})}$ was sub-relativistic, insufficient to drive forward Raman and self-modulation instabilities rapidly enough to produce a detectable wake.

Reviewer #1 comment 8: • Page 8, Discussion: The section starts with “The frequent appearance...” where a value for the relative frequency would support this statement.

Our response: This phrase referred back to a statement in the preceding paragraph that spectral peaks “were observed on approximately 90% of ~70 electron-yielding shots....” That said, we understand that papers are often not read linearly or completely. **Thus we took the reviewer's suggestion to repeat this information in the opening sentence of the Discussion, which now reads: “The appearance of quasi-monoenergetic collimated electron spectral features on ~90% of electron-yielding shots suggests...”**

Reviewer #1 comment 9: • Page 8, right column, end of first paragraph “... tolerances on pointing jitter and bunch compression.” I think that references to those values should be provided for transparency.

Our response: **In the revised manuscript we expanded the statement to include timing jitter:**

“...tolerances on pointing and timing jitter and bunch compression [56].”

and cited the paper [56] B. Marchetti *et al.*, “Conceptual and technical design aspects of accelerators for external injection in LWFA,” *Appl. Sci.* **8, 757 (2018), which discusses all of the tolerances in detail with extensive references to prior literature.**

Reviewer #1 comment 10: *Significance. The results shown are of high significance for laser-plasma electron acceleration. Due to long wavelengths and low densities, large “bubbles” (the accelerating structures) can be generated which allow for de-coupling of electron injection and acceleration by easing the accuracy requirements at the accelerating stage. In particular, low energy spreads are of high importance for applications. The chosen technology platform is currently not widely established for immediate application experiments or reproduction of the results. Yet the technology is available and can be implemented where needed. I do not see this as a showstopper for publishing the results.*

Our response: We appreciate this assessment of the significance of our results.

Response to Reviewer #2.

Reviewer #2 summary comment: *The manuscript describes experimental results, simulations, and analysis of wakewave excitation and electron acceleration experiments driven by ~10-um-wavelength, high-power (up to 3 TW), short-pulse (down to 2 ps), CO₂ laser. The results extend notably the driving laser wavelength used in such experiments till now: 0.8 um wavelength used routinely, 3.9 um wavelength laser used in Ref. [10]. With longer wavelength, the acceleration at rather low plasma densities (~10¹⁷ cm⁻³) can be driven by relatively low-power lasers (a few TW). Low densities open perspectives for potentially sub-mm accelerating structures: compared to several to tens-of-um structures used typically, sub-mm scale will allow easier coupling with linear accelerators and easier diagnostics. Importantly, driving laser in this work is a discharge-pumped gas laser, which is very different from the used-up-to-now solid-state lasers. This is a long-expected milestone for the high-power CPA CO₂ lasers, which opens several potential applications including those apart from compact electron accelerators.*

In addition to the key results summarized above, the authors explained several experimental observations and clarified the physics of laser-plasma interaction under the experimental conditions, namely excitation of the wakewave in the self-modulated (long-pulse: 4 ps) regime and estimated its threshold conditions and wave amplitude using sidebands of collective Thomson scattering; estimated the self-injection threshold; explained the role of relativistic self-focusing in the electron acceleration in the short-pulse (2 ps) regime. The authors analysed the electron spectra in the range of sub-MeV to 15 MeV: the spectra were mostly exponential but often contained quasi-monoenergetic features with a few percent relative bandwidth, and electron beam profiles, which exhibited divergent halo (typically 100 mrad) with presence of much lower-divergence, much brighter beamlets near the centre. The authors argued that these features corresponded to the onset of the forced wakefield regime, and that the strongly nonlinear bubble regime can be reached with a not-so-distant upgrade to ~20 TW, <1 ps, which would provide improvements in the electron beam quality.

The present version of the manuscript gives a clear overall understanding of what was done, however, there are several unclear places and figures, which I enumerate below. Also, the relevance of the simulations should be emphasized by comparing the initial setup and results with the experiment on the same graphs. There are also a few optional comments driven by my curiosity, however, I expect readers will have similar desires to know. Thus, why not to add a few (possibly supplementary) figures?

Our response: We appreciate this positive summary.

Reviewer #2, comment 1: *Page 1: "the momentum ratio $a_0 = e E_L / \omega_L m_e c$ ". The a_0 quantity is usually called dimensionless amplitude, why not to use this standard term?*

Our response: We took the reviewer's suggestion by adding the phrase in boldface type below:

"...i.e. the momentum ratio $a_0 = e E_L / \omega_L m_e c$, often called the dimensionless field amplitude or normalized vector potential, must exceed 1."

Reviewer #2, comment 2: *Page 2: "an off-axis parabola (OAP) mirror focused linearly-polarized drive pulses from ATF's CPA CO₂ laser [11, 12] to spot size $w_0 \sim 27.5$ um at the entrance of a supersonic hydrogen gas jet". Please clarify more precisely how far away from the gas jet centre the focal plane was located, or, even better, show it in Fig. 1(a); am I correct in my assumption that the focal plane in the simulation was located at the same position?*

Our response: We added the phrases in boldface type below to the quoted passage to provide the requested information:

"...an off-axis parabola (OAP) mirror focused linearly-polarized drive pulses from ATF's CPA CO₂ laser [11,12] to spot size $w_0 \sim 27.5$ μm at a focal plane located 1 ± 0.1 mm before the center of a supersonic hydrogen gas jet with an axially symmetric profile of 2 mm diameter. The dashed curve in Fig. 1(a) shows an idealized electron density profile $n_e(z)$ of the ionized gas jet along the laser

propagation axis, here with plateau density $n_e = 5 \times 10^{17} \text{ cm}^{-3}$, that we used for simulations. **In simulations, the vacuum laser focal plane was at $z = 0.1 \text{ mm}$, near the beginning of the density plateau (see Methods/Simulations for further discussion).**"

We used $z = 0.1 \text{ mm}$ to simulate both CTS and electron acceleration experiments. **A new paragraph at the end of Methods/Simulations** and our response to Comment #11 of Reviewer #3 explain the reasoning behind this compromise choice. Even though the focus was moved forward in electron acceleration experiments to a position that optimized electron yield and energy, simulations best reproduced these optimized results when we used a vacuum focus of $z = 0.1 \text{ mm}$. We attribute this discrepancy to the highly nonlinear nature of the interaction in acceleration experiments, which can amplify the importance of small deviations of jet and laser profiles (beyond measurement resolution) from the idealized shapes used in simulations. These simulations, this discrepancy notwithstanding, represent our best effort to reproduce the experimental results, which are the primary purpose of this report.

Reviewer #2, comment 3: Page 2, please correct the inline formula " $\lambda_p/2 = k_p/\pi$ " (should be π/k_p).

Our response: Thanks for the correction. **We fixed it.**

Reviewer #2, comment 4: Page 3: "Simulations of the interaction that include tunneling ionization, exemplified by Fig. 1(b)-(d), show that under these conditions, the wings of the focused CO_2 laser pulse self-ionize a hydrogen column of radius $R_p \sim 50 \text{ um}$ ": it is unclear if this is relevant to the experiment, because the ionization radius depends on the spot shape at low-intensity (far-from-the-peak) regions. For a Gaussian spot, the ionization radius would be relatively small, for a Lorentzian spot of the same Full Width at Half Maximum (FWHM), it would be larger, for a non-ideal spot due to aberrations, spatiotemporal couplings, etc., the ionization radius can be many times larger than for an ideal spot. (Please also see my related comments #19 and #24).

Our response: We agree that the exact value of the ionization radius depends on the spot shape. However, the exact radius is not relevant to the experiment. All that matters is that this radius exceeds λ_p over the n_e range of interest, so that the plasma supports a wake without edge effects, and without the need for pre-ionizing a wider plasma column. This was stated several sentences later:

The drive pulse generated wakes of transverse radius $R_w \approx w_0 \approx 20 \text{ um}$ that lay entirely within the self-ionized plasma column. Thus pre-ionization was not essential to produce a plasma wide enough to support the wake oscillations.

The simulated R_p for a Gaussian pulse thus serves as a lower limit for the actual R_p . If it is wider, then all the better for the experiment. **We revised the quoted passage as follows to clarify these issues:**

Simulations of the interaction that include tunneling ionization, exemplified by Fig. 1(b)-(d), show that under these conditions, the wings of a focused **Gaussian** CO_2 laser pulse self-ionize a hydrogen column of radius $R_p \sim 50 \text{ um}$ [see Fig. 1(b)]. **Non-Gaussian (e.g. Lorentzian, aberrated) pulses of the same FWHM would ionize an even wider column because of their more intense wings. Regardless of the exact R_p ,** the drive pulse generated wakes of transverse radius $R_w \approx w_0 \approx 20 \text{ um}$ that lay entirely within the self-ionized plasma column. Thus pre-ionization was not essential to produce a plasma wide enough to support the wake oscillations.

Reviewer #2, comment 5: The authors discuss the self-focusing threshold up to the middle of the first paragraph on page 3, and then abruptly use the self-injection threshold at the end of this paragraph and in Fig. 1. Inattentive readers like me will miss this difference and, like me, will be greatly confused. Thus, 0.5 TW is above self-focusing threshold for $n_e = 5 \times 10^{17} \text{ cm}^{-3}$, but below the self-injection threshold. Please explain and possibly discuss this for the benefit of the readers.

Our response: Thank you for this feedback. **In the revised manuscript, we added the passages in boldface type below to alert readers that we were switching from self-focusing threshold to self-injection threshold:**

Even though wakes formed for $P_L > P_{cr}$, the $p_z(x, z)$ profile in Fig. 1(c) shows only electron momenta attributable to the wake oscillations themselves. The additional z-momentum that would be expected if the wake had trapped and accelerated electrons is not present. This means that at $P_L = 0.5$ TW and $N_e = 5 \times 10^{17}$ cm⁻³ we are above the self-focusing threshold needed to form wake structures, but below the self-injection threshold. This absence of self-injected, trapped electrons accelerating along the wake propagation direction in these simulations corroborates our observation of no accelerated electrons.

Reviewer #2, comment 6: Page 3: "Our simulations showed a strong influence of the dynamic ionization model on the structure and evolution of the SM wakes compared to the pre-ionized plasma approximation [27]. Likewise, including the ion motion in simulation leads to the formation of ionization channels (Fig. 1 and [24]) which are largely suppressed in simulations using the fixed ion approximation [27]." These statements represent results that are not the most important for this work but still of interest. Unfortunately, they are not supported by figures. It would be nice to add comparison with pre-ionized and stationary-ion cases, probably as supplementary figures.

Our response: The comparative results were published in Refs. [24] and [27] cited in the quoted passage, where they are supported by figures. To make this clearer, **we changed the quoted passage to read:**

Our simulations **presented in [27]** showed a strong influence of the dynamic ionization model on the structure and evolution of the SM wakes compared to the pre-ionized plasma approximation. **These results show that in dynamically ionized gas the laser pulse modulates more strongly, and stronger wakes form earlier because of the stronger ponderomotive force.** Likewise, including ion motion leads to the formation of ion channels (Fig. 1(d) and [24]), which are largely suppressed in simulations using the fixed ion approximation (see results in [27]). **Fig. 1(d) and Fig. s3 (Supplementary Material) directly compare wakes formed with moving and stationary ions, respectively.**

Reviewer #2, comment 7: Fig. 1(c,e), the colour scales showing the electron momenta are not clear. The electric field is possible to see, although even this could be improved (for example, blue is a part of the momentum colour scale, as well as colour of the field curve). It might be better not to overlap the momenta and fields, although the authors may use other methods to clarify the figure.

Our response: **We revised Fig. 1 to present field curves in black, a color that is not included in the momentum color scale. In addition we adjusted the color scale so that low- p_z electrons, formerly rendered in barely visible light yellow, now appear more prominently as bright yellow/orange. We also used 3D particle data (instead of 2D slices, which missed some electrons) to enumerate self-injected/accelerated electrons, so that they show up more clearly in panel (e). Finally, we added panel (f), which presents the spatial and energy distributions of electrons accelerated by the wake in Fig. 1d after they have propagated out of the interaction region into vacuum. This clearly displays accelerated electron population without overlapping wakefield data.**

Reviewer #2, comment 8: As far as I understand, Fig. 1(d,e) shows simulations with conditions close to the experiment with 2 TW pulses where the electrons were accelerated, Figs. 6 and 7. Please compare simulated and experimental electron spectra and divergences/beam profiles.

Our response: **We added a simulated beam profile as panel c) of Fig. 6 and a simulated electron spectrum to panel b5) of Fig. 7, where they can be compared directly with data.** Simulations results correspond to the electron bunch from 5×10^{17} cm⁻³, accelerated by 2 TW laser pulses.

Reviewer #2, comment 9: Page 4: "Through such measurements we confirmed, as shown in Fig. 3(b), that sideband intensity within this window, was vanishingly weak or absent for $n_e < 4 \times 10^{17} \text{ cm}^{-3}$ for excitation at P_L . From Eq. 1, this "turn-on" density corresponds to $P_{cr} = P_L = 0.5 \text{ TW}$." However, the laser intensity is relativistic even without self-focusing, $a_0 \sim 2.2$, as written at the bottom of page 2. Thus, I would expect some wakewave excitation anyway, accompanied by corresponding measurable sidebands. What is wrong in my logic?

Our response: First of all, as Reviewer #3 (Comment #2) correctly pointed out, the a_0 value cited at the bottom of page 2 was incorrectly calculated --- it is actually 1.8. After being corrected on that one, we checked that other a_0 values throughout the paper were all correct. Secondly, the main logical error is that "some wakewave excitation" is not equivalent to "measurable sidebands". As discussed at length in the paragraph following the quoted passage:

The main factors governing sideband intensity are: *i*) wake amplitude δn_e ; *ii*) wake lifetime within the 4 ps probe longitudinal envelope; *iii*) wake location within that envelope; *iv*) dephasing between co-propagating wake and probe. Generally, strong sidebands occur when a high-amplitude wake filling a large fraction of the most intense portion of probe's longitudinal envelope co-propagates with the probe for approximately one coherence length [31].

The data in Fig. 3(b) is telling us, not that wakes are not excited at all at $n_e = 0.4 \times 10^{18} \text{ cm}^{-3}$, only that they do not generate "measurable sidebands" in view of one or more of factors *i*) thru *iv*). That is why we worded the quoted passage to say that "sideband intensity" (not "wake amplitude") was "vanishingly weak or absent" for the stated conditions.

The reviewer could (and should) ask: why does sideband intensity become "vanishingly weak" also for $n_e > 1.0 \times 10^{18} \text{ cm}^{-3}$, as also clearly shown in Fig. 3(b)? At $n_e = 1.0 \times 10^{18} \text{ cm}^{-3}$, $P_{cr} = 0.1 \text{ TW}$, so $P_L/P_{cr} = 5$ and vacuum a_0 is still 1.8, but will increase in the plasma because of self-focusing. These are even more compelling conditions for wake excitation than $n_e = 0.4 \times 10^{18} \text{ cm}^{-3}$, $P_L/P_{cr} = 1$. Nevertheless, as also explained qualitatively in the paragraph following the quoted passage, factors *i*) thru *iv*) conspire to suppress sideband intensity:

Eventually, however, the pump over-focused, generating wakes that decayed and became chaotic faster, and formed earlier both within the probe profile, where intensity was weaker, and within the jet, resulting in stronger dephasing. These factors combined to weaken CTS.

The simulation results superposed on the data in Fig. 3(b) support the picture described above. **To summarize the main point, we added the following reminder at the end of the paragraph following the quoted passage:**

These results emphasize that sideband intensity is not simply proportional to wake amplitude.

Reviewer #2, comment 10: Fig. 2 caption: " n_e -independent sum- and difference-frequency peaks". However, it is known that the laser pulse frequency gradually downshifts due to energy transfer to plasma waves, at least in the bubble regime [Nie et al., Nature Communications 11 2787 (2020)], and this downshift is density-dependent. Can the authors check if this effect can be observed, and describe its absence (not measurable within ???nm detection threshold) or presence (above ???nm detection threshold)?

Our response: The downshifts observed by Nie et al. can be thought of as originating from the sharp plasma-density downramp (and thus refractive-index upramp) that the most intense part of the drive pulse experiences as a result of overlapping and co-propagating with the front edge of the bubble. In the self-modulated regime, in contrast, the drive pulse envelope overlaps dozens of plasma periods, and thus overlaps dozens of plasma-density downramps (at the front of each bucket) and upramps (at the back of each bucket). These downramps and upramps induce red-shifts and blue-shifts in different parts of the drive pulse envelope, leaving its center frequency unchanged. Similar red-shifts and blue-shifts are responsible for the density-dependent Stokes and anti-Stokes sidebands that we observe on our co-

propagating probe pulse (see Fig. 2). Since the drive pulse is doing work on the plasma, its Stokes (red-shifted) sideband must be stronger than its anti-Stokes (blue-shifted) sideband. This sideband asymmetry is the analog in the self-modulated regime of the whole-pump downshift that Nie et al. observe in the bubble regime. However, these sidebands are far too weak, compared to the pump's unshifted central peak, to generate observable difference-frequency signals of their own. The theoretical expectation is therefore that the pump-probe difference-frequency signal frequency should be density-independent, as correctly stated in the paper. The reviewer has incorrectly extrapolated a signature of pump depletion from the bubble regime that does not exist in the same form in the self-modulated regime.

Reviewer #2, comment 11: Fig. 4 caption: "pump-probe difference-frequency peaks at ~560 nm appear at all P_L ." This contradicts text on page 5: "DFG signals were observed at normalized power down to $P_L/P_{cr} \sim 0.5$."

Our response: Thanks for pointing this out. We changed the passage in the Fig. 4 caption to read:

"pump-probe difference-frequency peaks at ~560 nm are detectable down to $P_L \sim 0.5 P_{cr}$."

Reviewer #2, comment 12: Fig. 5: the simulated curve shown in (d) should be overlapped with the experimental curve shown in (c), and the apparent difference in their widths explained or at least discussed. Also, the pump-probe cross-correlation curve in Fig. 5(c) can be scaled to have its peak similar to the data points; that would reduce apparent discrepancy in temporal width.

Our response: We took the reviewer's suggestions to superpose the simulated data in (d), convolved with the pump-probe cross-correlation as discussed below, on the data in Fig. 5(c) [see green dashed curve], and to scale up the cross-correlation curve in Fig. 5(c). We added the following passage to the paragraph describing Fig. 5 to explain the new presentation:

Fig.5(d) shows results of a SPACE simulation that included a *continuous-wave* (CW) probe beam ($\lambda_{pr} = 2.5\mu\text{m}$) co-propagating with the $10\mu\text{m}$, 4 ps drive pulse, shown by the red dashed curve. Using a CW probe saved computational time, because the time response could be deduced in a single computational run from the CTS response at different locations within the probe. The ~1ps relaxation is consistent with the dephasing time of the inhomogeneously-broadened Raman-shifted lines. The resulting plot [data points and blue connecting lines in Fig. 5(d)], however, is not convolved with the 4 ps probe used in the experiments. Including this probe convolution yields the dotted green curve in panel c). It is slightly broader than the pump-probe autocorrelation, shown by the red-dashed curve in panel c), but slightly narrower than the measured response. The simulation could not capture the weaker, more slowly-decaying CTS signal because of numerical noise. Its ~ 30 ps relaxation is consistent with the decay time of electron plasma waves into ion acoustic waves [48].

Reviewer #2, comment 13: Fig. 6(b): it would be nice to colour the data points according to the density, similar to (c).

Our response: We took the reviewer's suggestion to color data points according to density. In response to Comment #6 of Reviewer 1, the original Fig. 6(b) is merged with the original Fig. 6(a). The combined plot is now called Fig. 6(a).

Reviewer #2, comment 14: Page 7: "The threshold power P_L for generating detectable charge was now only $0.3P_{cr}$, more than 3x smaller than the wake formation threshold observed by CTS for longer, less powerful pulses [Fig. 1(d)-(e)]. Such a low appearance threshold indicates that self-focusing was no longer the principal mechanism triggering wake formation or acceleration. To help understand this threshold,

Fig. 6(c) re-plots data in Fig. 6(b) as a function of vacuum focused a_0 for three fixed n_e . Now there are three n_e -dependent appearance thresholds at $a_0(\text{vac}) = 1.9$ (for $n_e = 1.6e17 \text{ cm}^{-3}$), 2.4 (for $n_e = 0.9e17 \text{ cm}^{-3}$), and 3.4 for $n_e = 0.4e17 \text{ cm}^{-3}$." However, looking at the single trend on Fig. 6(b), and three different trends (different for each n_e) on Fig. 6(c), I would guess that P/P_{crit} ratio is more important than a_0 , meaning that the self-focusing is still the dominant governing process. Although, the authors are correct in emphasising that the value $P/P_{\text{crit}} \sim 0.3$, rather than ~ 1 , is a bit surprising.

Our response: Now that we have implemented the reviewer's suggestion in Comment #13 above, the "single trend" in the former Fig. 6(b) [now part of the merged Fig. 6(a)] is no longer so clear. The lowest density ($n_e = 0.4 \times 10^{17} \text{ cm}^{-3}$) contributes only a single data point, and the second-lowest ($n_e = 0.9 \times 10^{17} \text{ cm}^{-3}$) contributes data points only over the limited range ($0.55 < P/P_{\text{cr}} < 1.4$). In neither case is the exact electron "appearance threshold" clearly evident. Moreover, we added about a dozen data points for the $n_e = 0.9 \times 10^{17} \text{ cm}^{-3}$ case that were not included in the original plot, and found that half of them fell well below the "single trend". For the two higher densities ($n_e = 1.4$ and $4.0 \times 10^{17} \text{ cm}^{-3}$), there are many more data points over a wider range of P/P_{cr} , making the appearance thresholds clear. But they are different. So on closer inspection, the data do not support the existence of a "single trend" strongly enough to support the conclusion that self-focusing is the dominant mechanism.

Moreover, in our judgment it is not only "a bit surprising" but unphysical for self-focusing to be "the dominant governing process" when $P < P_{\text{cr}}$. The simple fact that we are generating wakes and electrons for P/P_{cr} as small as 0.3 by itself demonstrates that self-focusing cannot be the dominant mechanism. Thus we stand by our original interpretation that relativistic vacuum a_0 is the principal operative mechanism in this regime, and thank the reviewer for helping us to present the evidence in a more consistent manner.

See our response to Reviewer #1, Comments #6 and #7 for further relevant discussion.

Reviewer #2, comment 15: Page 7: "an intense core with a divergence as small as $\sim 10 \text{ mrad}$." However, I do not see 10 mrad divergence in Figs. 6(d-f); it may be due to the colour scale used. It would be nice to show (linear-scale) lineouts on top of the images - that would help to reveal the small-divergence core the authors write about, and discriminate it from the $\sim 100 \text{ mrad}$ diffuse background.

Our response: In the revised manuscript, we re-worded the quoted passage to read:

...an intense core with divergence $40 \gtrsim \Delta\theta_{\text{HWHM}} \gtrsim 12 \text{ mrad}$.

Then, to help the reader see the smallest of these, added...

Black arrows on Fig. 6(d) [to] highlight one of the narrowest such peaks observed, with $\Delta\theta_{\text{HWHM}} \approx 12 \text{ mrad}$.

Reviewer #2, comment 16: Page 7: "Fig. 7(a2) displays an exponential distribution, possibly the high-energy tail of a distribution similar to the one shown in Fig. 7(a1)." It would be of interest to know the temperatures of the exponential electron spectra.

Our response: We fit the energy distribution curves in Figs. 7(b1) and 7(b2) to Boltzmann functions $\exp(-E_e/k_B T_e)$. The revised figure caption describes them as follows:

Black dots in panels b1 and b2 are fits of the exponential tail of the energy distribution curve to functions $\exp(-E_e/k_B T_e)$, with $k_B T_e = 1.2$ (0.62) MeV for panel b1 (b2).

We also added the following sentence to the text describing Fig. 7:

Black dots in panels (b1) and (b2) show fits of the high-energy tail of these energy distributions to exponential functions $\exp(-E_e/k_B T_e)$, where k_B is Boltzmann's constant and T_e is an effective electron temperature.

Reviewer #2, comment 17: Page 8: "When de-convolved from the spectrometer's instrumental width due to its 2mm entrance slit, this corresponds to fractional energy spread $dE_e/E_e \sim 0.01$." However, it seems that this electron beamlet divergence is small enough to pass thru the slit without clipping, and thus the slit itself should not affect the recorded energy width (the slit affects the absolute energy error bar, actually - please show it, by the way). On the other hand, the electron beamlet divergence in the horizontal direction does affect the apparent energy width. A deconvolution can be performed assuming the same horizontal and vertical divergence, although I am not sure if such assumption can be justified in this particular case: a supporting fact in favour of this hypothesis is that "The angular width of the quasi-monoenergetic peaks scaled approximately in proportion to its energy width," mentioned below. The same comment applies to the data shown in Fig. 7(a4,b4).

Our response: The reviewer points out correctly that this beamlet's divergence (cone angle $\Delta\theta_{\text{HWHM}} \approx 2$ mrad) is small enough to pass thru the slit (acceptance cone angle ~ 9 mrad) w/o clipping. Thus "the spectrometer's instrumental width due to its 2mm entrance slit" indeed does not limit energy resolution. What *does* limit energy resolution in this case is the beamlet's inherent $\Delta\theta_{\text{HWHM}} \approx 2$ mrad divergence, determined from the "vertical" spread of the recorded beamlet feature, unimpeded by the slit. On the reasonable assumption* that "horizontal" and "vertical" divergences are equal, the former is responsible for about 2/3 of the horizontal width of the recorded beamlet feature. This was actually the basis for the "deconvolution" of a 0.03 fractional energy bandwidth to infer a 0.01 fractional width. We stand by that conclusion. We regret incorrectly describing the deconvolution procedure, and are grateful to the reviewer for catching the error.

*Spectrally-integrated luminescence images such as those shown in Fig. 6(d)-(f) show no evidence of a consistent difference between "vertical" and "horizontal" beam divergence. Thus the assumption that they are equal is reasonable.

We changed the quoted passage to reflect the points above:

...this beamlet's inherent divergence is responsible for $\sim 2/3$ of the horizontal width of the recorded feature. When divergence is deconvolved, we obtain fractional energy spread $\Delta E_e/E_e \approx 0.01$ (i.e. 7.7 ± 0.07 MeV),...

Reviewer #2, comment 18: Page 8 Summary: "Accordingly our fully 3D PIC simulations have accurately modeled four key observables: (1) the SM-wake excitation threshold $P_L \sim P_{cr}$; (2) the self-injection threshold $a_0 \sim 3$; (3) the dependence of wake amplitude on n_e at fixed P_L and Δt , and on (d) Δt at fixed P_L and n_e ." However, (1) requires some explanation, see my point #5 above; (2) this threshold is not discussed in the text and is not obvious from the figures; (3) it is not obvious for me where this dependence is shown: is it Fig. 3(b)? but in this case, this is dependence of the CTS intensity vs. density; of course, CTS intensity is connected with the wake amplitude, as described in the text, but they are not identical; (d) - probably, this should read (4) - again, I do not see the dependence of the wake amplitude on Δt , only the normalized sideband amplitude on Δt in Fig. 5(d).

Our response: Regarding (1), the SM-wake excitation threshold: **we provided the explanation in response to point #5. We revised the quoted passage as follows to address these concerns:**

Accordingly our fully 3D PIC simulations have accurately modeled four key observables: (1) the SM-wake excitation threshold $P_L \approx P_{cr}$ for $n_e > 4 \times 10^{17} \text{ cm}^{-3}$ [see Fig. 1(b)-(c)]; (2) the onset of self-injection for wakes excited by 2 TW pulses [see Fig. 1(d)-(f)]; (3) the dependence of CTS sideband intensity (which is related to wake amplitude) on n_e at fixed P_L and Δt [see Fig. 3(b)] and on (4) Δt at fixed P_L and n_e [see Fig. 4(a)].

Reviewer #2, comment 19: Methods: CO₂ laser: the measured focal spot, pulse shape, and spectrum, and their typical shot-to-shot and day-to-day variations should be shown, for ~ 0.5 TW and ~ 2 TW laser modes. Also, in the text it was mentioned that the spot quality degraded at higher laser powers: both "good" and

"degraded" spots should be shown. This is necessary because propagation and self-focusing of a laser pulse in low-density plasma is strongly affected by the spot and pulse shapes. Without these figures and discussion, the results cannot be reproduced.

Our response: We added Figures 8, 9, and 10 showing data on the focal spot, spectrum, and pulse duration, respectively, along with two paragraphs of new text (highlighted in red type in the red-lined version of the revised manuscript) to the subsection Methods/CO₂ laser. The new information includes comparisons of 0.5 TW and 2 TW configurations. We cite Ref. [12] for additional information.

In view of the exceptionally stable focal profile shown by this data, we changed the text description of "degraded" high-intensity focus (Results/Generation of self-modulated wakes/paragraph 3) to "slightly degraded":

Occasional individual pulses with τ_L as small as 1.8 ps, \mathcal{E}_L as high as 6 J, and P_L approaching ~3 TW, albeit with slightly degraded focus (see Methods), were measured at the vacuum interaction region.

Reviewer #2, comment 20: Methods: CO₂ laser: please mention the repetition rate used in the experiment. Is it determined by the laser or experimental setup?

Our response: We added the following passage to Methods/CO₂ laser:

The laser operates in a single-shot regime. The minimum time delay between shots, dictated by the capacitor charging time of the Marx generator powering the discharge in the final amplifier, is approximately 20 seconds. However, we deliberately extend this interval to at least 1.5 minutes between shots to allow adequate cooldown of the spark gaps (high voltage switching components). This practice helps to prolong their operational lifetime.

We also added a brief description of potential future upgrades:

We are also in the initial stages of research and development aimed at increasing the laser repetition rate to 1 Hz and beyond. Currently, we are of the view that this increase would require a shift from electric-discharge to optical pumping methods [51].

We added the new reference [51] Tovey et al., Appl. Opt. 58, 5756 (2019).

Reviewer #2, comment 21: Methods: Probe laser: "These signals were sent to phase-locked-loops, which compared them to the RF reference signal and fed back error signals to the piezo-mirrors, which in turn adjusted the mirror positions to maintain fixed relative phases." As far as I understand, the intended meaning of this sentence is that the pump and probe laser oscillators were both synchronised to the same RF wave, and the "fixed relative phases" are relative to this wave. However, it is ambiguous, because my first impression that the pump and probe lasers were synchronised such that their relative phases were fixed, implying sub-cycle accuracy. I suggest to re-word the description to avoid misunderstanding.

Our response: We removed the misleading phrase "fixed relative phases" from the quoted passage, and re-worded it as follows:

These signals were sent to phase-locked-loops, which compared them to the RF reference signal and fed back error signals to the piezo-mirrors. This feedback adjusted the mirror positions to ensure phase locking of the pump and probe laser oscillators with the RF reference signal.

Reviewer #2, comment 22: Methods: Probe laser: it would be nice to show a figure showing the scan determining the delay-zero timing; if it was published somewhere, a reference would do. What is the delay-zero timing accuracy?

Our response: We added the following statement to 2nd paragraph of Methods/Probe laser:

The electronic synchronization loops at ATF are designed to operate with sub-picosecond timing jitter, a level that is negligible under the conditions of this work. The best evidence for this is that the rise-time of pump-probe data in our experiments [e.g. Fig.4(c), blue data points and curve] closely matched the pump-probe cross-correlation (dashed red curve) determined from single-shot autocorrelation measurements.

ATF's policy is to operate the system at a synchronization level that experiments require, as determined by user measurements such as this one. They do not invest resources in further optimizing timing jitter until an experiment demands it. They refrain from publishing or circulating internal system measurements beyond those that users themselves obtain, or that are previously published.

Reviewer #2, comment 23: *Methods: Accelerated electron characterization: please describe if the actual 3D magnetic field distributions were measured, or nominal fields and sizes were used in electron spectrometer energy calibration.*

Our response: **We added the following passage to Methods/Accelerated electron characterization to address this question:**

The cited magnetic fields and their associated fringe fields were profiled with a Hall probe, and the measured fields used in calculating electron trajectories through the spectrometer. The 9 mrad cone angle of electrons entering the magnetic fields was, however, the main factor limiting energy resolution to 10%.

Reviewer #2, comment 24: *Methods: Simulations: Please add plasma-free (vacuum) spot and pulse shapes used in simulations to the figures showing measured spot and pulse, for comparison. The self-focused spot shapes would also be of interest.*

Our response: **The inset of the new Fig. 8 (added in response to Comment #19) directly compares the measured vacuum spot size and shape with those used in simulations. The caption of Fig. 8 describes this inset as follows:**

Inset: Comparison of averaged measured radial profile of 4 J pulse profile from (a) (open circles) with Gaussian profile of $w_0 = 27.5 \mu\text{m}$ (dark red curve) used in simulations.

A new paragraph added at the end of Methods/Simulations (mentioned in our response to Comment #2), states:

Laser pulse profile: **A Gaussian vacuum spatial (temporal) profile with $w_0 = 27.5 \mu\text{m}$ ($\tau_L = 4$ or 2 ps) was used because it closely matched vacuum focus spot profile (autocorrelation) measurements [Fig. 8(c), inset; Fig. 10 and Ref. (12)]. These measurements provided no clear guidance for exploring deviations from Gaussian profiles.**

Intensity autocorrelation measurements do not unambiguously determine the longitudinal pulse profile. An assumed Gaussian profile fit measured autocorrelation traces reasonably well. Ref. (12) reports refinements to a Gaussian profile based on a theoretical model of the CO₂ gain medium that yielded slightly better fits on test shots, but the required refinements varied with radius and from shot to shot, and would have required impractical inline autocorrelation measurements to implement. We concluded that there was negligible added value to labor-intensive efforts to improve on a Gaussian spatio-temporal profile model.

Reviewer #2, comment 25: *Methods or main text: please describe how the gas jet density profile was obtained. The present description on page 4, "calibrated by independent optical measurements of the ionized gas jet," is too brief and does not reveal the method. What is the density error bar?*

Our response: **We expanded the quoted passage to read as follows:**

...calibrated by independent optical measurements of the density profile $n_e(z)$ of the ionized gas jet with $\pm 10\%$ accuracy using an ionization-induced plasma grating technique [34].

We added the new reference [34] Zhang *et al.*, *Plasma Phys. Control. Fusion* 63, 095011 (2021), which describes the technique and its error sources in detail.

Response to Reviewer #3.

Reviewer #3 summary comment: *This is an important work, it shows electron trapping and acceleration with a long wavelength laser driver in a plasma accelerator for the first time. This was achieved by having a high peak power laser as well as close to transverse matching to the plasma. Simulations of the high peak power CO2 laser pulses demonstrate that wavebreaking is expected for the upgraded laser case only however, a plasma wave is excited in both cases via the self-modulation instability. Detailed measurements of a laser probing the interaction which scattered off the plasma presented which allowed for an estimate of the plasma wave amplitude. Finally, the accelerated electron beam was characterised with an investigation into the conditions that trigger injection in the transition between self-modulated and blow out regimes. A large volume of data was collected over a broad range of parameter space and its analysis is well communicated to the reader. This allows for a high degree of confidence in the trends and conclusions. The paper is well written and method of analysis sound. Below follow comments and suggestions for further improvement.*

Our response: We appreciate this positive assessment.

Reviewer #3, Comment 1: *Abstract gives neutral gas backing pressures of target in atmospheres however, the remainder of the paper discusses plasma electron number density. Since demonstrating wave excitation at low density is a key outcome of the paper, I think that it would be easier to read if the abstract was consistent with the remainder of the paper.*

Our response: We revised the statements in question as follows:

...in hydrogen plasma of **electron density down to $4 \times 10^{17} \text{ cm}^{-3}$** (~1/100 atmospheric density)...

... down to **$3 \times 10^{16} \text{ cm}^{-3}$**

We used "atmospheres" in the Abstract to connect better with the wide readership of *Nature Communications*. Everyone knows what an "atmosphere" is, whereas readers outside the physical sciences may not readily connect a particle density of " 10^{17} cm^{-3} " with their everyday experience. The above is a compromise.

Reviewer #3, Comment 2: *The calculation of normalised laser intensity should be checked. I could not reproduce $a_0 \approx 2.2$ for 2J, 4 ps, $\lambda = 10.3 \text{ }\mu\text{m}$ and $\omega_0 = 27.5 \text{ }\mu\text{m}$. Instead, I get $a_0 \approx 1.8$.*

Our response: Thanks for this correction. We agree with the reviewer's number, **and have corrected the manuscript accordingly**. We double-checked that other a_0 values cited in the manuscript are correct.

Reviewer #3, Comment 3: *For the laser intensity with the upgraded laser, it should be made clearer that this is the peak intensity assuming the highest overserved laser pulse energy, not the typical pulse energy. In this case, I can reproduce the intensity given in the text.*

Our response: **We added the phrase in boldface type to clarify this point:**

Vacuum peak intensity now reached $I_0 \approx 2.5 \times 10^{17} \text{ W/cm}^2$ ($a_0 \approx 3.9$) for **6 J pulses**.

We also revised the calculated a_0 value downward slightly from 4.1 to **3.9**. The original number was obtained using wavelength $\lambda = 10 \text{ }\mu\text{m}$, whereas the actual wavelength for the 2 ps laser configuration is $9.2 \text{ }\mu\text{m}$.

Reviewer #3, Comment 4: *In the text and figure 1, it is shown that electron bunches are accelerated by the 2 TW laser but no other information is given other than the maximum electron energy. What is the energy distribution of these electrons? How does it compare to the experimental measurements?*

Our response: See our response to Reviewer #2, Comment 8. **We added the simulated energy distribution of these electrons to panel b5) of Fig. 7, where it is compared to experimental measurements.**

Reviewer #3, Comment 5: *In figures 3(b) and (c) the sideband intensity could be normalised to better compare where the peaks occur between the time delay and Stokes peaks to cross section respectively.*

Our response: Figure 3 has no panel (c). From the mention of "time delay" we guess that the reviewer may be referring to Fig. 5. We do not understand to what standard the reviewer is suggesting that we "normalize" the sideband intensity. In principle one could normalize it to the peak probe intensity. However, we do not have enough information to do so accurately or meaningfully. The unshifted probe signal is blocked, so a direct comparison from the data presented is not available. Moreover, it is very difficult to calibrate the energy of each sideband in absolute units because it depends on transmission through the spectrometer slit, grating and detector efficiencies, pump-probe spatial overlap, etc. Even if such a normalization could be carried out, it would be difficult to interpret in a meaningful way because of the major role of wave vector mismatch in determining sideband intensity (see Eq. 2 and accompanying text).

In addition, we have puzzled over the phrase "...to better compare where peaks occur between the time delay and Stokes peaks to cross section respectively", which evidently is supposed to motivate the request for normalization. We have been unable to make any sense of this phrase.

Since we understand neither what is being requested, nor why, nor even for certain which Figure is meant, we have chosen not to respond further to this comment.

Reviewer #3, Comment 6: *Figure 4(a) is difficult to interpret, particularly the trend in red.*

Our response: We regret that this reviewer, unlike the other two, found this panel difficult to interpret. Unfortunately, the comment provides no guidance on the reason for the difficulty.

We speculated that it may have stemmed from the thickness of the dashed red line at $P/P_{cr} = 1$, which at first glance could have been mistaken as a continuation of the solid red curve. **In the revised manuscript we therefore thinned the red dashed line to match the other dashed red line in the same figure.** Hopefully that clarifies the figure.

We are happy to revise the figure, or any text referring to it, if there are additional concerns. But first we need to understand what those concerns are.

Reviewer #3, Comment 7: *An explanation should be given for the multiple injection events from the interaction. Do the simulations give an insight?*

Our response: The phrase "multiple injection events" is imprecise. It could refer to: 1) injection into multiple buckets of the wake; 2) continuous injection into any one bucket at multiple locations along the wake's propagation through the plasma; 3) injection into multiple locations within any one bucket. In any case, our response is that "multiple injection" in all 3 senses is the norm for self-modulated LWFA. In contrast to the quasi-static bubble-regime, SM-LWFA relies on an instability which does not lend itself to micro-management. In this regime, the drive pulse covers multiple buckets, so injection occurs more or less equivalently into all of them. Holistic "wave-breaking", rather than a localized convergence of bubble

sheath currents at the back of a singly-driven bubble, triggers injection, spilling electrons randomly through each bucket. And since wave growth and wave breaking are the culmination of an instability, there is no stopping them once they start. Clever "self-truncation", "downramp", or "colliding pulse" techniques used to localize injection into plasma bubbles are ineffective. Yes, our simulations show all of these effects, as well as "multiple injection" in all 3 senses. They are inherent features of the SM regime.

To explain the multiple injection inherent to SM-LWFA physics, we added the bold-faced passages below to the end of the sub-section "Results/Generation of self-modulated wakes":

...In contrast to the $p_z(x,z)$ profile in Fig. 1(c), copious electrons with relativistic p_z are now evident. In Fig. 1(e), electron bunches accelerated to $p_z/m_e c \sim 40$ are distributed among the **multiple** accelerating bins of the wake **that the drive pulse overlapped**. Moreover **they are distributed randomly throughout each bin because** the plasma wave had broken, injecting plasma electrons **at uncontrolled initial locations and times prior to their trapping in the** wake's accelerating potential. **This wave-breaking and injection, once started in mid-jet, continued through the end of the interaction, since they are the culmination of the forward Raman instability.**

Reviewer #3, Comment 8: A statement should be given as to whether the simulations also seen the narrow energy spread beam within the overall accelerated electron population.

Our response: We added the following statement to the text, just prior to the Discussion section:

This comparison shows that simulations reproduce the peaked (as opposed to exponential) spectra observed on the high-energy screen with close to the observed peak energy, but do not capture the exceptionally narrow features shown in rows 3 and 4 of Fig. 7.

See also our response to Comment #4.

Reviewer #3, Comment 9: A comment should be made on the pointing stability of the electron beam. How many shots containing a narrow divergence beam are expected to overlap with the position of the entrance slit of the energy spectrometer magnet?

Our response: We added a sentence to the end of the 4th paragraph of the Results/Measurements of accelerated electrons section stating:

The centroids of these narrow beamlets exhibited shot-to-shot RMS pointing fluctuations of ~ 5 mrad.

This is consistent with $\sim 90\%$ of such features entering a 2mm spectrometers slit with a 9 mrad acceptance cone angle, and thus with the observation stated in the submitted manuscript:

...spectral peaks [were] observed on approximately 90% of ~ 70 electron-yielding shots characterized with the spectrometer in its higher energy configuration.

Reviewer #3, Comment 10: It is stated that the pulse duration varied "significantly" from shot-to-shot, was the pulse duration measured on shot to correctly determine the peak power of that shot?

Our response: The statement in question (Methods/CO₂ laser) actually states that "pulse duration varied insignificantly from shot-to-shot...". Because of an unfortunately placed line break, the prefix "in-" occurred on the line preceding "significantly". See our response to Reviewer #2, Comment 19 for additional detail added to Methods regarding pulse duration measurements. Because pulse duration varied insignificantly from shot to shot, we relied on spot checks instead of on-shot measurements.

Reviewer #3, Comment 11: *In the simulations, what was the position of the (vacuum) laser focus? Experimentally, the laser was focused at both the beginning and centre of the gas jet. Were the simulations performed with a static simulation window?*

Our response: All simulations were performed with a static window in the laboratory frame, since we wish to observe the structure and evolution of wakes in a spatially extended domain, unlike the moving window computational approach which focuses on a small domain near the laser front. **We added a phrase to Methods/Simulations to state this.**

Regarding the vacuum laser focus, see our response to Comment #2 of Reviewer #2. The focus location in all simulations was at $z = 100\mu\text{m}$ [referenced to the horizontal scale of Fig. 1(a)]. First of all, on reflection we stated the "center-of-jet" location of the vacuum focus for electron acceleration experiments with too much precision in the original manuscript. **In the revised manuscript, we softened the language as follows:**

Electron yield peaked with the vacuum laser focus shifted forward toward the center, rather than exactly at the entrance, of the gas jet. We adjusted the exact vacuum focus location empirically with each run to maximize yield, but generally it lay between the entrance and center for experiments.

We added the following passage to Methods/Simulations:

Test simulations were carried out to optimize gas jet shape, laser pulse profile, and vacuum laser focus location used in presented simulation runs. These tests balanced four criteria: 1) best match to known experimental conditions, within measurement uncertainty; 2) best match to experimental results (sideband intensities, electron yield and spectrum); 3) best simulation efficiency; 4) least sensitivity to small changes in input parameters. Simulations of CTS experiments [Fig. 1(b),(c); Fig. 3(b); Fig. 5(d)] came close to meeting all four criteria simultaneously. Simulations of electron acceleration experiments [Fig. 1(d)-(f); Fig. 6(c); Fig. 7(b5)] required some compromise. We made the following decisions:

...

- **Vacuum laser focus position:** In test simulations of CTS experiments, results were insensitive to small changes in vacuum focus location over the range $0 < z < 0.1$ mm [referring to the horizontal scale of Fig. 1(a)]. In test simulations of electron acceleration experiments, for which the vacuum focus was shifted toward the gas jet center ($z = 1$ mm), results best matched experimental electron yield and energy for $z = 0.1$ mm. Thus, as a compromise, we used a vacuum focus at $z = 0.1$ mm for simulations of all experiments. This is smaller than the average z used in electron acceleration experiments. We attribute the discrepancy tentatively to the highly nonlinear nature of the interaction in those experiments, as a result of which small deviations of gas jet and laser pulse profiles from their ideal shapes can play an outsized role in the results of the experiment.

Reviewer #1 (Remarks to the Author):

The revision has significantly increased the clarity of the manuscript. I have no further objections regarding publishing the manuscript as is.

Reviewer #2 (Remarks to the Author):

Reviewer: Alexander S. Pirozhkov

The authors significantly improved the manuscript and answered satisfactory to my questions. I can now recommend the publication. A few points remain which need further clarifications or corrections, please see below.

1. Page 7: there are two separated sentences within the same paragraph: "Nevertheless a weak component of both sidebands persisted out to $\Delta t \sim 30$ ps, suggesting a slower decay mechanism for low-amplitude wakes." and "Its ~ 30 ps relaxation is consistent with the decay time of electron plasma waves into ion acoustic waves [48]." which, it seems, discuss the same 30 ps tail. If this is so, it would be reasonable to combine them or at least make them closer together.
2. Methods, page 10: "Fig. 6 of Ref. [12] shows typical autocorrelations traces and temporal pulse profiles in the 2 ps, 2TW configuration..." Fig. 6b of Ref. [12] contains a post-pulse at ~ 27 ps. I believe that similar post-pulse could be present during the experiments described in this manuscript. Thus, it can be an alternative explanation of the sidebands stretching up to 30 ps. This alternative explanation should be mentioned or explicitly rejected with justification.
3. Yet another explanation could be a 20-30 ps tail or prepulse due to imperfect compensation of the 3-rd order dispersion, which is difficult to characterize (and therefore remove) with autocorrelators.
4. The presentation of the experimental data on the new version of Fig. 6 is now clearer. However, one thing is still confusing: The discussion of Fig. 6(a) on Page 7: "Remaining data points show yield for shots of higher power ($2 < P_L < 3.3$ TW) driving less dense plasma: $n_e = 1.6 \times 10^{17} \text{ cm}^{-3}$ ($P_{cr} = 1.5$ TW, inverted black triangles);". Thus, I would expect to see data within the range of $2 \text{ TW}/1.5 \text{ TW} < P_L/P_{cr} < 3.3 \text{ TW}/1.5 \text{ TW}$, i.e. $1.3 < P_L/P_{cr} < 2.2$. However, Fig. 6(a) shows the inverted black triangles in the range of $\sim 0.2 < P_L/P_{cr} < 2$. The upper value is consistent, but the lower value, $P_L/P_{cr} \sim 0.2$, corresponds to ~ 0.3 TW, which is much lower than the laser power used ($2 < P_L < 3.3$ TW). Please clarify. Also, without these black points, the appearance thresholds would be $P_L \sim (0.7 \pm 0.1) P_{cr}$ for all other cases. Thus, the discussion of the appearance thresholds may also be reconsidered appropriately.
5. Fig. 6(c-f): it would be nice to add the linear-scale vertical and horizontal lineouts across the electron distribution peaks, as I suggested in the previous review round.
6. The text on Page 7: "Electron yield peaked with the vacuum laser focus shifted forward toward the center, rather than exactly at the entrance, of the gas jet. We adjusted the exact vacuum focus location empirically with each run to maximize yield, but generally it lay between the entrance and center for experiments." and Methods Page 11 "We characterized accelerated electrons generated with the vacuum laser focus at the center of the gas jet, where we observed maximum electron yield." are inconsistent. Please make appropriate corrections.

Key to type: 11 pt. italic type: reviewer comments

11 pt. standard type: our responses to Rev. 2 (magenta)

Boldface type: description of manuscript changes in response to comments

Reference numbers in responses below are those of the revised manuscript.

2nd Response to Reviewer #2.

Reviewer #2, summary comment: *The authors significantly improved the manuscript and answered satisfactory to my questions. I can now recommend the publication. A few points remain which need further clarifications or corrections, please see below.*

Our response: Thanks for helping us to improve the manuscript.

Reviewer #2, comment 1: *Page 7: there are two separated sentences within the same paragraph: "Nevertheless a weak component of both sidebands persisted out to $\Delta t \sim 30$ ps, suggesting a slower decay mechanism for low-amplitude wakes." and "Its ~ 30 ps relaxation is consistent with the decay time of electron plasma waves into ion acoustic waves [48]." which, it seems, discuss the same 30 ps tail. If this is so, it would be reasonable to combine them or at least make them closer together.*

Our response: We combined the two sentences as suggested.

Reviewer #2, comment 2: *Methods, page 10: "Fig. 6 of Ref. [12] shows typical autocorrelations traces and temporal pulse profiles in the 2 ps, 2TW configuration..." Fig. 6b of Ref. [12] contains a post-pulse at ~ 27 ps. I believe that similar post-pulse could be present during the experiments described in this manuscript. Thus, it can be an alternative explanation of the sidebands stretching up to 30 ps. This alternative explanation should be mentioned or explicitly rejected with justification.*

Our response: We added the following sentence to the main text to reject this possibility explicitly:

We have ruled out the possibility that postpulses caused this delayed feature since autocorrelation measurements [12] do not detect any postpulses in the interval $0 \leq \Delta t \leq 25$ ps, and those detected at longer Δt are too weak to generate wakefields detectable by CTS.

A postpulse at $\Delta t \approx 27$ ps would generate a secondary wake starting at this delay. It could not explain a gradually decaying signal in the interval $0 \leq \Delta t \leq 25$ ps as observed. Moreover, as stated in Ref. [12], the main pulse contains 87% of the total measured energy. Throughout this work, we did not detect any wakes when the 4 ps main pulse was attenuated to $< 13\%$ of its maximum energy of 2 J.

Reviewer #2, comment 3: *Yet another explanation could be a 20-30 ps tail or prepulse due to imperfect compensation of the 3-rd order dispersion, which is difficult to characterize (and therefore remove) with autocorrelators.*

Our response: No such pedestal was every measured. Moreover it is unlikely because, in contrast to typical CPA configurations for solid-state lasers, where the pulse is stretched from femtoseconds to a nanosecond, our configuration involves stretching a 2-ps pulse to only ~ 80 ps.

Reviewer #2, comment 4: *The presentation of the experimental data on the new version of Fig. 6 is now clearer. However, one thing is still confusing: The discussion of Fig. 6(a) on Page 7: "Remaining data points show yield for shots of higher power ($2 < P_L < 3.3$ TW) driving less dense plasma: $n_e = 1.6 \times 10^{17} \text{ cm}^{-3}$ ($P_{cr} = 1.5$ TW, inverted black triangles);". Thus, I would expect to see data within the range of $2 \text{ TW}/1.5 \text{ TW} < P_L/P_{cr} < 3.3 \text{ TW}/1.5 \text{ TW}$, i.e. $1.3 < P_L/P_{cr} < 2.2$. However, Fig. 6(a) shows the inverted black triangles in the range of $\sim 0.2 < P_L/P_{cr} < 2$. The upper value is consistent, but the lower*

value, $P_L/P_{cr} \sim 0.2$, corresponds to ~ 0.3 TW, which is much lower than the laser power used ($2 < P_L < 3.3$ TW). Please clarify. Also, without these black points, the appearance thresholds would be $P_L \sim (0.7 \pm 0.1)P_{cr}$ for all other cases. Thus, the discussion of the appearance thresholds may also be reconsidered appropriately.

Our response: Thanks for catching this inconsistency. The power range stated parenthetically is incorrect. The lower limit of 2 should have been 0.3, as the reviewer correctly points out. The intended meaning was that the remaining data *includes* shots of higher power than in the preceding data driving less dense plasma, but it indeed also includes lower powers. All of the data in Fig. 6a are correctly plotted. We corrected the statement in question as indicated below:

Remaining data points show yield for shots **up to 3.3 TW peak power** driving less dense plasma...

Reviewer #2, comment 5: Fig. 6(c-f): it would be nice to add the linear-scale vertical and horizontal lineouts across the electron distribution peaks, as I suggested in the previous review round.

Our response: **We added the lineouts as suggested.**

Reviewer #2, comment 6: The text on Page 7: "Electron yield peaked with the vacuum laser focus shifted forward toward the center, rather than exactly at the entrance, of the gas jet. We adjusted the exact vacuum focus location empirically with each run to maximize yield, but generally it lay between the entrance and center for experiments." and Methods Page 11 "We characterized accelerated electrons generated with the vacuum laser focus at the center of the gas jet, where we observed maximum electron yield." are inconsistent. Please make appropriate corrections.

Our response: Thanks for pointing out this inconsistency. The first statement is correct. **We corrected the second one to be consistent with it.**

Reviewer #2 (Remarks to the Author):

Reviewer: Alexander S. Pirozhkov

The authors answered my questions and made the required corrections, I recommend publication of the manuscript.